# Hypothermia Shifts Neurodegeneration Phenotype in Neonatal Human Hypoxic–Ischemic Encephalopathy but Not in Related Piglet Models: Possible Relationship to Toxic Conformer and Intrinsically Disordered Prion-like Protein Accumulation

**DOI:** 10.3390/cells14080586

**Published:** 2025-04-12

**Authors:** Lee J. Martin, Jennifer K. Lee, Mark V. Niedzwiecki, Adriana Amrein Almira, Cameron Javdan, May W. Chen, Valerie Olberding, Stephen M. Brown, Dongseok Park, Sophie Yohannan, Hasitha Putcha, Becky Zheng, Annalise Garrido, Jordan Benderoth, Chloe Kisner, Javid Ghaemmaghami, Frances J. Northington, Panagiotis Kratimenos

**Affiliations:** 1Department of Pathology, Division of Neuropathology, Johns Hopkins University School of Medicine, 558 Ross Building, 720 Rutland Avenue, Baltimore, MD 20205-2196, USA; dpark58@jhmi.edu (D.P.); bzheng15@jh.edu (B.Z.);; 2Department of Neuroscience, Johns Hopkins University School of Medicine, 558 Ross Building, 720 Rutland Avenue, Baltimore, MD 20205-2196, USA; 3Department of Anesthesiology and Critical Care Medicine, Johns Hopkins University School of Medicine, 558 Ross Building, 720 Rutland Avenue, Baltimore, MD 20205-2196, USA; 4The Pathobiology Graduate Training Program, Johns Hopkins University School of Medicine, 558 Ross Building, 720 Rutland Avenue, Baltimore, MD 20205-2196, USA; 5Department of Pediatrics, Johns Hopkins University School of Medicine, CMSC, 600 North Wolfe Street, Baltimore, MD 21287-0001, USA; 6Department of Pediatrics, Children’s National Hospital, George Washington University School of Medicine and Health Sciences, Washington, DC 20010-2916, USA

**Keywords:** cell death, connectome, neonatal brain injury, oligodendrocyte, α-synuclein, prion protein, nuclear SOD1, toxic oligomer

## Abstract

Hypothermia (HT) is used clinically for neonatal hypoxic–ischemic encephalopathy (HIE); however, the brain protection is incomplete and selective regional vulnerability and lifelong consequences remain. Refractory damage and impairment with HT cooling/rewarming could result from unchecked or altered persisting cell death and proteinopathy. We tested two hypotheses: (1) HT modifies neurodegeneration type, and (2) intrinsically disordered proteins (IDPs) and encephalopathy cause toxic conformer protein (TCP) proteinopathy neonatally. We studied postmortem human neonatal HIE cases with or without therapeutic HT, neonatal piglets subjected to global hypoxia-ischemia (HI) with and without HT or combinations of HI and quinolinic acid (QA) excitotoxicity surviving for 29–96 h to 14 days, and human oligodendrocytes and neurons exposed to QA for cell models. In human and piglet encephalopathies with normothermia, the neuropathology by hematoxylin and eosin staining was similar; necrotic cell degeneration predominated. With HT, neurodegeneration morphology shifted to apoptosis-necrosis hybrid and apoptotic forms in human HIE, while neurons in HI piglets were unshifting and protected robustly. Oligomers and putative TCPs of α-synuclein (αSyn), nitrated-Syn and aggregated αSyn, misfolded/oxidized superoxide dismutase-1 (SOD1), and prion protein (PrP) were detected with highly specific antibodies by immunohistochemistry, immunofluorescence, and immunoblotting. αSyn and SOD1 TCPs were seen in human HIE brains regardless of HT treatment. αSyn and SOD1 TCPs were detected as early as 29 h after injury in piglets and QA-injured human oligodendrocytes and neurons in culture. Cell immunophenotyping by immunofluorescence showed αSyn detected with antibodies to aggregated/oligomerized protein; nitrated-Syn accumulated in neurons, sometimes appearing as focal dendritic aggregations. Co-localization also showed aberrant αSyn accumulating in presynaptic terminals. Proteinase K-resistant PrP accumulated in ischemic Purkinje cells, and their target regions had PrP-positive neuritic plaque-like pathology. Immunofluorescence revealed misfolded/oxidized SOD1 in neurons, axons, astrocytes, and oligodendrocytes. HT attenuated TCP formation in piglets. We conclude that HT differentially affects brain damage in humans and piglets. HT shifts neuronal cell death to other forms in human while blocking ischemic necrosis in piglet for sustained protection. HI and excitotoxicity also acutely induce formation of TCPs and prion-like proteins from IDPs globally throughout the brain in gray matter and white matter. HT attenuates proteinopathy in piglets but seemingly not in humans. Shifting of cell death type and aberrant toxic protein formation could explain the selective system vulnerability, connectome spreading, and persistent damage seen in neonatal HIE leading to lifelong consequences even after HT treatment.

## 1. Introduction

Perinatal asphyxia is a global health problem resulting in ~23% of the deaths of neonates worldwide [1]. Neonatal asphyxia causes hypoxic–ischemic encephalopathy (HIE) that evolves spatiotemporally within selective brain regions and neural systems [2,3,4,5,6]. Primary sensory and forebrain motor regions are keenly vulnerable [4,7]. White matter is also exquisitely sensitive to perinatal hypoxia-ischemia (HI) [5]. In Western countries, therapeutic hypothermia (HT) is the standard of care for moderate to severe HIE in term infants because it can improve outcomes for survival or neurodevelopmental impairment of some children at 18 months of age [8,9]. However, the benefits of HT are time-sensitive and have limited holistic efficacy long-term [10]. For every seven infants treated with HT, only one infant will avoid death or moderate to severe disability compared to infants maintained at normothermia (NT) [8]. This ratio for effect can be adjusted depending on the specific outcome measured [11,12]. Therefore, many surviving HIE infants can have considerable morbidity and challenging lifelong consequences, including movement disability and impairments in learning, memory, vision, emotional regulation, and executive functioning [10].

The clinical limitations of HT in treating neonatal HIE [10] are not understood, vis-à-vis the robust protection seen in animal models of neonatal HI [13,14,15,16,17,18]. Potential explanations could be discovered in the cellular and molecular pathology of human HIE. Possible contributing explanations include differences in neuronal cell death mechanisms and dendritic biology in human and animals [19,20], multiorgan failure [21], and the comorbidity of seizures [18]. White matter, with its extensive elaboration in the commissural and association tracts in human brain [22], could be unresponsive or even adversely affected by brain cooling [23,24]. Excitotoxicity, mitochondrial failure, and oxidative stress are implicated in the acute primary neurodegeneration in human HIE and animal models of HI [25,26,27,28]. Delayed neuropathology with its apparent network distribution putatively involves secondary metabolic failure and regional connectivity perturbations [7,25,28,29,30]. Both acute and delayed neurodegeneration may shift along the apoptosis–necrosis neuronal cell death continuum in the presence of HT [31]. Neuronal and glial cell degeneration in cooled and rewarmed individuals could also be influenced by unresolving proteinopathy stemming from injury-related and thermal-sensitive protein oxidation and folding [32,33,34] and changes in protein translation, stability, and degradation [35,36,37,38].

Human neurodegenerative diseases of the mature central nervous system (CNS) often show sentinel discrete injury and incipient focal clinical presentation that progress, sometimes rapidly and aggressively, with a preferential distribution of neuropathology within the connectome [31,39,40,41]. Amyotrophic lateral sclerosis (ALS), Parkinson’s disease (PD), Alzheimer’s disease (AD), frontotemporal dementia, multiple system atrophy (MSA), and Creutzfeldt–Jakob disease (CJD) all show apparent trans-synaptic propagation of neuropathology throughout the CNS with varying tempos of clinical disability and fatal outcome [31,39,41,42]. Proteinopathy might drive these diseases, involving intrinsically disordered proteins (IDPs), protein aggregates and oligomers, and toxic conformer proteins (TCPs) [42,43,44,45,46]. The canonical TCP is prion protein (PrP) [47], but other proteins can adopt PrP-like properties and show pathological intercellular and trans-synaptic spreading. For example, α-synuclein (αSyn) and TAR DNA binding protein-43 have IDP characteristics; superoxide dismutase-1 (SOD1), amyloid β protein, and tau protein can become TCPs [42,43,44,45,46,47,48]. In addition to seeding and intercellular propagation, intracellular accumulation of aberrant forms of these proteins can cause proteasome dysfunction leading to cell death [49,50,51].

Perinatal brain injuries seen clinically and in animal models have similarities with many adult neurological diseases. Commonalities are connectivity-related evolution of neuropathology, white matter axon and oligodendrocyte vulnerability, oxidative stress and proteinopathy, proteasome dysfunction, mitochondrial pathobiology, and cell death types that manifest along the cell death continuum [28,30,31,32,52,53,54]. Despite the pathological and putative mechanistic similarities along the infant-adult brain injury/disease continuum, the clinical application of HT (targeted temperature management) is used mostly for moderate to severe neonatal HIE. In this study, we used postmortem neonatal human HIE brains and neonatal piglets with encephalopathies to test the hypothesis that neurodegeneration type is modified by HT treatment; we then examined, after using human oligodendrocyte and neuron cell culture for proof-of-concept, if IDP and TCP proteinopathies occur globally in the context of injury in the developing brain, as they do in the adult brain. Of particular interest was synucleinopathy, defined here as the antibody-based detection, with immunohistochemistry, immunofluorescence, or Western blot, of α-synuclein (α-Syn) protein in abnormal states involving post-translational modification (nitration) and forms of the protein different from the monomeric form (aggregated or oligomerized) implying an altered function or a pathological manifestation, including a toxic gain-of-function [55,56,57,58].

## 2. Materials and Methods

### 2.1. Human Autopsy Brain Samples

All experiments were designed and reported in accordance with ARRIVE guidelines. Postmortem human brain samples were obtained from the Johns Hopkins University School of Medicine Brain Resource Center (Division of Neuropathology, Department of Pathology), the National Institute of Child Health and Development (NICHD) Brain and Tissue Bank for Developmental Disorders at the University of Maryland, Baltimore, MD, and the Children’s National Hospital, Washington, DC. All autopsies had approved consent and tissues were de-identified. The protocols for using human autopsy tissue were reviewed and approved by the JHMI-IRB (NO: 02-09024-04e, approved 27 September 2002) and the Children’s National Hospital IRB (IRB#15350, approved 22 December 2020, and #11850, approved 17 January 2019). The human postmortem cases used are summarized in Table 1. The distant archival HIE cases predated HT as the standard of care, so these cases were normothermic. More recent HIE cases received HT protocols prior to death or missed the therapeutic opportunity for HT. Infantile human non-HIE cases of spinal muscular atrophy (SMA) or acute deaths due to non-neurological causes such as accidental death, pneumonia, or drug intoxication were used as comparators.

All human brains were immersion fixed in neutral-buffered formalin for at least 2 weeks before brain cutting. Paraffin-embedded brain sample blocks (Appendix A) were cut on a rotary microtome into 5- to 10-µm sections. The sections were mounted on glass slides for hematoxylin and eosin (H&E) staining, immunohistochemistry (IHC), and immunofluorescence. Brain samples of human ALS and PD or their transgenic mouse models [31,59,60,61] were used as proteinopathy-positive controls for IHC and Western blot.

### 2.2. Neonatal Piglet Brain Injury and Survival Models

The animal protocols were reviewed and approved on 6 June 2023 by the Institutional Animal Use and Care Committee of Johns Hopkins University (protocol number SW23M119). Neonatal Yorkshire piglets (2 to 4 days old, weighing 1–2 kg, males) were subjected to one of three different brain injury protocols (Table 2) with survivals of 29–96 h as described [18,32,62,63,64] or 14 days. The study design used males only to limit the number of animals needed and because we have not seen sex differences in our piglet models. The different piglet protocols were used because each had specific advantages and justifications (Table 2).

Our standard, randomized piglet HI model has been described [62]. Anesthesia was induced with isoflurane 5% and 50% nitrous oxide in 50% oxygen delivered by a nose cone. After intubation, the anesthetic was changed to 1.5–2% isoflurane and 70% nitrous oxide in 30% oxygen. Sterile catheters were placed in the external jugular vein and femoral artery. A fentanyl bolus (20 µg/kg iv) was given, followed by 20 µg/kg/h. Additional fentanyl (10–20 µg/kg boluses) was given as needed for discomfort management. After intubation and vascular line catheterization, inhaled O_2_ was decreased to 10% for 45 min. Then, after 5 min of room air, required for successful cardiac resuscitation in this model, the piglets were made asphyxic for 8 min by clamping the endotracheal tube to cause severe bradycardia and hypotension, followed by cardiopulmonary resuscitation (CPR) using 50% inhaled oxygen, manual chest compressions, and epinephrine (100 µg/kg iv). The inhaled oxygen was decreased to 30% after return of spontaneous circulation. Recovery from anesthesia, wake-up, and extubation occurred ~3 h later. Sham piglets received anesthesia, surgery, and inhaled 30% oxygen, but without hypoxia or cardiac arrest. After recovery, blood gas analyses demonstrated normal O_2_ saturations, so the piglets were not chronically hypoxic. These piglets survived for 4 days (Table 2), and the brains were prepared with fixation for IHC and immunofluorescence (*n* = 6 sham and *n* = 6 HI) or without fixation for Western blot (*n* = 6 sham and *n* = 6 HI). The brains were prepared blinded to treatment.

The piglet HI-HT protocol (Table 2) has also been described [18,32,54]. Piglets were randomized into one of four experimental groups: sham normothermia (NT), sham HT, HI-NT, or HI-HT. A non-anesthetized, not operated cohort of piglets was a naïve control group. The brain insult followed the standard HI protocol described above. Whole body HT was initiated two hours after CPR using ice packs and a cooling blanket to a rectal temperature of 34 °C. Rewarming (0.5 °C/h) to NT began in HT piglets at 20 h from the onset by increasing the temperature of the water circulating through the blanket. Piglets reached their NT target temperature of 38.5 °C/h at ~29 h from onset. Piglets were sacrificed at 29 h to obtain unfixed freshly frozen brain tissue for Western blot with group sizes of sham-NT (*n* = 4), sham-HT (*n* = 4), HI-NT (*n* = 4), HI-HT (*n* = 4), and naïve (*n* = 4). Other piglets were perfusion fixed at 2–7 days for H&E staining to determine neuroprotection with the group sizes being sham-NT (*n* = 6), sham-HT (*n* = 10), HI-NT (*n* = 8), HI-HT (*n* = 10), and naïve (*n* = 6).

Other piglets were used in a novel 2-hit brain injury protocol that combines the HI from asphyxic cardiac arrest and excitotoxicity caused by intracerebral stereotaxic microinjection of the *N*-methyl-*D*-aspartate (NMDA) glutamate receptor agonist quinolinic acid (QA). In this HI-QA brain injury protocol, the anesthesia, sterile catheter placement, and hypoxia protocols were like those described. Asphyxia was induced similarly, but only for 6 min. With this protocol, piglets develop bradycardia and hypotension with heart rates <60 beats per minute or mean arterial blood pressure (MAP) less than 45 mmHg. The piglets were resuscitated similarly as described above. After a 30 min stabilization period, the piglet head was installed into a stereotaxic frame (Kopf Instruments, Tujunga, CA, USA) in a flat skull position. Skull burr holes were made under sterile surgery and used to deliver, by micromanipulator and microsyringe, intracerebral injections of QA into the right somatosensory cortex (8 µL, 480 mmol) and left lateral geniculate nucleus (10 µL, 600 nmol). Stereotaxic coordinates were selected roughly from the atlas of Salinas-Zeballos et al. [65] with precise reproducible targeting established empirically for the 2–3-day old piglet [18,66]. These targets were chosen to focally ablate and unilaterally inactivate somatosensory and visual networks, respectively, for separate fMRI BOLD studies. These pigs survived for 48 h with normal blood gases (free of hypoxia) and were used for immunoblotting: sham (*n* = 4), vehicle (*n* = 4), naive (*n* = 4), and HI-QA (*n* = 4); or the piglets were perfusion fixed at 14 days, and the brains were used for histology/IHC/immunofluorescence. Control piglet groups were vehicle injected, sham procedure, and naive.

### 2.3. Piglet Brain Harvesting

The triaging of the piglet models and their different survivals are explained (Table 2). All piglets received a lethal dose of pentobarbital 50 mg/kg and phenytoin 6.4 mg/kg (SomnaSol, Dublin, OH, USA) for euthanasia and brain harvesting. For unfixed samples, after thoracotomy and left myocardial puncture and aortic catheterization, ice-cold 100 mM phosphate-buffered saline (PBS, pH 7.4) was perfused (~2 L) for body exsanguination. The brain was removed quickly from the skull and then cut into slabs and regionally microdissected on a metal plate chilled on wet ice [67]. Brain samples were snap frozen in isopentane cooled in dry ice. All samples were stored in individual Eppendorf tubes at −80 °C until used. For histological endpoints, after PBS exsanguination, the piglets were perfused with freshly prepared 4% paraformaldehyde (PF) in 100 mM phosphate buffer (pH 7.4) for brain fixation (~4 L). After perfusion, the animal was decapitated, and the head was immersed in 4% PF solution for in situ fixation overnight. This was performed to prevent neuropathological artifacts during brain removal prematurely from the skull. The next day, the brain was removed from the skull base and again placed in 4% PF solution for overnight. Then, each brain, left and right cerebral hemispheres, was blocked in the coronal plane from the frontal lobes to the hindbrain, including the telencephalon, diencephalon, midbrain, pons, and medulla with cerebellum, and paraffin processed in tissue cassettes [18]. The paraffin-embedded brain blocks were cut on a rotary microtome into 10 µm sections and mounted on glass slides for H&E staining, IHC, and immunofluorescence as described [18].

### 2.4. Human iPS Cell- and Embryonic Stem Cell-Derived Neural Cell Models of QA Excitotoxicity and TCP Proteinopathy

To complement the human infant autopsy brain and neonatal piglet neuropathology descriptions and to provide proof-of-concept, we developed human neural cell type-specific models to test the hypothesis of rapid TCP formation after excitotoxic injury. Oligodendrocyte and neuronal cell lineages were used because these cells are vulnerable in the HI injured developing brain [26]. For oligodendrocytes, we used a human fibroblast induced-pluripotent stem (iPS) cell-derived neural progenitor cell line (Tempo Bioscience, San Francisco, CA, USA). This non-oncogenically immortalized cell line was generated using integration-free episomal technology to induce pluripotency. For neurons, we used the embryonic stem cell H9 line that can be directed to differentiate exquisitely into functional neurons [19]. The cells were cultured under atmospheric oxygen with 5% CO_2_ and differentiated and maintained for ~20–30 days. The cultures were never subjected to hypoxia.

Oligoprogenitor cells were plated on Matrigel and propagated in DMEM-F12 media containing 2 mM glutamine, non-essential amino acids, StemPro neural supplement, 10 ng/mL human platelet-derived growth factor (PDGF), 10 ng/mL human neurotrophin-3 (NT3), 100 ng/mL biotin, and 5 µM cyclic adenosine monophosphate (cAMP). Accutase was used for passaging. Directed differentiation to oligodendrocyte progenitor cells was achieved in DMEM-F12/Neurobasal (50:50) media supplemented with B27, 2 mM glutamine, non-essential amino acids, 100 ng/mL biotin, 1 µM cAMP, 20 µg/mL ascorbic acid, 10 ng/mL human brain-derived neurotrophic factor (BDNF), and 200 ng/mL triiodothyronine (T3). Induction of differentiation in this media with T3 was essential for proper oligodendrocyte differentiation for 12–21 days. At 24 h after plating, 10 µM cytosine arabinofuranoside (AraC) was added to the media for 24 h to inhibit cell proliferation, then new media was used without AraC.

Neuronal differentiation of H9 cells was induced as described [19]. Neurosphere-derived neural precursors were plated on polyornithine/laminin-coated plates with neurobasal complete medium in the absence of mitogens. Then, they were cultured in differentiation medium (DMEM/F12 supplement, 1% N2 supplement) containing 1% B27 supplement, 1 mM cAMP, 20 ng/mL brain-derived neurotrophic factor, and 20 ng/mL glial cell line-derived neurotrophic factor.

Live human oligodendrocytes and neurons cells were imaged to assess morphological differentiation and then were fixed in 4% PF with 20% sucrose for cell phenotyping by immunofluorescence. The human neuron culture has been characterized in detail [19]. Our human oligodendrocyte culture has not been characterized before and is performed so here. This was performed using antibodies to PDGF receptor-α (PDGFRα, mouse monoclonal IgG, clone JF104-6, Novus Biologicals, Centennial, CO, USA), oligodendrocyte marker O4 (mouse monoclonal IgM, clone O4, R&D Systems, Minneapolis, MN, USA), 2’,3’-cyclic nucleotide 3’-phosphodiesterase (CNPase, mouse monoclonal IgG, clone 11.5B, Chemicom-Millipore, St. Louis, MO, USA), Olig2 (rabbit polyclonal IgG, Chemicon-Millipore), bridging integrator-1 (BIN1, rabbit polyclonal IgG, Proteintech, Rosemont, IL, USA), class 3 β-tubulin (mouse monoclonal IgG, clone TUJ1, Covance, Dedham, MA, USA) and glial fibrillary acidic protein (GFAP, rabbit polyclonal IgG, Dako, Santa Clara, CA, USA).

NMDA glutamate receptor excitotoxicity was used as an injury paradigm for our human neural cell cultures because this mechanism is believed to be relevant to human neonatal HIE [26], it is used in our neonatal piglet models of encephalopathy [66,67,68], and we find it more direct, reliable, and reproducible than oxygen/glucose deprivation. The cells were subjected to injury during a period of incomplete differentiation (mid-differentiation) by changing medium to DMEM-F12/Neurobasal without B27 and adding QA to the media at a final concentration of 20–100 µM. The cells were exposed to QA for 12 h and then were changed to complete differentiation media without QA. After 12 additional hours of culture, the cells were lysed and frozen for Western blot of TCPs, or they were fixed in PF solution for quantification of cell injury as assessed by cytoplasmic vacuolar pathology and apoptotic morphology.

### 2.5. Western Blotting

Immunoblotting was performed using highly specific antibodies to examine immunoreactive protein levels of pathological conformers of αSyn (nitrated-αSyn and aggregated-αSyn), misfolded/oxidized SOD1, and prion protein (PrP) (Table 3) in different brain regions and cell cultures. Frozen tissue samples were homogenized with a Brinkmann polytron in ice-cold 20 mM Tris HCl (pH 7.4) containing 10% (wt/vol) sucrose, 200 mM mannitol, complete protease inhibitor cocktail (Roche, Indianapolis, IN, USA), 0.1 mM phenylmethylsulfonyl fluoride, 10 mM benzamidine, 1 mM EDTA, and 5 mM EGTA. Crude homogenates and cell lysates were sonicated for 15 s and then centrifuged at 1000 g_av_ for 10 min (4 °C). Protein concentrations were measured by bicinchoninic acid assay with a kit (Pierce, Thermo Scientific, Carlsbad, CA, USA) using bovine serum albumin as a standard.

Piglet brain fractions and cell lysates were subjected to sodium dodecyl sulfate polyacrylamide gel electrophoresis (SDS-PAGE) and transferred to nitrocellulose membrane by electroelution [29,32,54]. Ponceau S staining of nitrocellulose membranes before immunoblotting verified the lane equivalency of sample loading and transfer in each experiment. Blots of crude tissue and cell lysates were blocked with 2.5% nonfat dry milk with 0.1% Tween 20 in 50 mM Tris-buffered saline (pH 7.4), then incubated overnight at 4 °C with primary antibody (Table 3). After the primary antibody incubation, blots were rinsed and then incubated with horseradish peroxidase-conjugated secondary antibody (0.2 µg/mL). For all blots, the primary and secondary antibodies were used at concentrations for visualizing protein immunoreactivity within the linear range. The blots were developed with enhanced chemiluminescence (Pierce) and imaged with a ChemiDoc imaging system (Bio-Rad, Hercules, CA, USA).

### 2.6. IHC and Immunofluorescence

Immunoperoxidase IHC, with diaminobenzidine (DAB) as chromogen, was performed on paraffin sections of human and piglet brain for single antigen labeling, as described, using sodium citrate for antigen retrieval and formic acid pretreatment [63,68,77,78]. Sections were stained with antibodies to nitrated-Syn, aggregated-αSyn, misfolded/oxidized SOD1, and PrP (Table 3). These antibodies have been characterized and validated and appear highly specific based on work by us and others (Table 3). With PrP immunostaining, proteinase K pretreatment (40 µg/mL in 100 mM phosphate buffer for 30 min at room temperature) of piglet brain sections was performed to determine the resistance of the immunoreactivity patterns to enzymatic digestion. This test can be used to distinguish the normal isosequential cellular PrP from the misfolded TCP prion-like form of PrP [79,80]. With single labeled sections, Nissl counterstaining with cresyl violet (CV) was performed for regional, laminar, and cellular identifications and for profile counting of total neurons.

We used double-labeling immunofluorescence and IHC for the identification of cell types and synapses harboring TCPs. Species origin-distinct primary antibodies to neurons, axons, presynaptic terminals, oligodendrocytes, and astrocytes were used (Table 4) in combination with antibodies to TCPs (Table 3). For immunofluorescence, sections were treated with TrueBlack autofluorescence quencher (Biotium); visualization of antigens in brain sections was performed using highly specific AlexaFluor Texas Red or 488 goat anti-rabbit or goat anti-mouse IgG (Invitrogen) at dilutions of 1:400. Section cover-slipping was performed using Vectashield Vibrance antifade mounting medium with or without DAPI (Vector Laboratories). Imaging was performed using a Zeiss Axiophot epifluorescence microscope and CaptaVision software v.Plus. Dual antigen detection was also performed with DAB, cobalt/nickel-DAB, or benzidine dihydrochloride (BDHC) as chromogens. The double antigen localizations with these chromogens yield nicely contrasted colors under brightfield microscopy for single- and double-labeled cell identification [27,81]. CV counterstaining was not performed for immunohistochemical double-labeled sections.

### 2.7. Quantification of H&E Neuropathology

A profile counting based approach was used for quantification of the amount of neuronal damage in H&E-stained human and piglet brain sections. At least three different 5 or 10 µm thick sections were analyzed for each brain region for each human or piglet brain. All counts were performed at 1000× magnification in random nonoverlapping microscopic fields (at least 20 microscopic fields per region per section) with careful z-axis focusing using an Olympus BH2 microscope and CaptaVision software. This high magnification was used so that nuclear features (chromatin clumping) and cytoplasmic vacuolation could be best discerned at light microscopic resolution. Split cells without a clear cut through the nucleus were excluded. For each different brain region per case about 600–1050 individual neurons were classified.

In human HIE and non-HIE brain samples, individual neuron profiles were counted in three different sections (per autopsy case) of frontal cortex, hippocampus (CA1 and CA4), and basal ganglia (caudate nucleus and putamen). These brain areas were chosen for counting because of the impression of very reliable regional sampling at autopsy and because of the relative pristine quality of the histology for postmortem tissue. Other areas of human neocortex, including temporal cortex and occipital cortex (Appendix A), were evaluated qualitatively for neuropathology, but not counted, because of uncertainties about the neuroanatomical sample matching on a case-by-case basis. Moreover, cortical layer definition was often obscured in some neocortical regions of the human HIE brains because of the tissue edema (Appendix A). The more superficial layers of the human HIE frontal cortex were the most reliably identifiable. In piglet brain samples from the HT/NT treatments, individual neuron profiles were counted in precisely matched levels of the frontal cortex (specifically motor cortex) and basal ganglia (striatal putamen). The frontal cortex and putamen counting in piglet treated as NT or HT was performed to match these regions in human HIE-NT and HIE-HT cases. The identification of these regions in piglet brain has been shown [18]. Unlike human HIE, the hippocampus was not counted in piglet brain H&E sections because it is variably affected after HI with large variation; damage largely depends on the co-morbidity of clinical and electroencephalographic seizures [7,18].

Counted neuronal profiles in H&E-stained sections of the human brain [59,78,81] and piglet brain [7,24,62,63,68,82] were classified by their microscopic appearances as described previously. This H&E classification cannot distinguish between interneurons and projection neurons and attritional degenerating interneurons from glia. Specifically, immunophenotyped subtypes of interneurons will be the topic of another study. These morphologies for cell degeneration have been shown before to be resolvable by H&E staining with the nuclear and cytoplasmic appearances weighted prominently in the divisibility. Normal neuron cell bodies (Figure 1A,B) were 8–30 μm in diameter with a non-vacuolated cytoplasm, interpreted as intact membranous organelles without swelling, and an open nucleus (not condensed, darkly basophilic, or pyknotic) with at least one nucleolus and chromatin strands dispersed in a finely particulate nucleoplasmic matrix. The ischemic-necrotic neuron in human and piglet brain had a hematoxylin (blue-purple)-stained, angular, and pyknotic nucleus, angular soma, vacuolated and eosinophilic (red-pink) cytoplasm, and absence of perinuclear pallor (Figure 1C,D). These cells undergo dissolution of the plasma and nuclear membranes (Figure 1C,D) and nucleoplasmic matrix speckling [7,62,66]. Cells undergoing the apoptosis–necrosis continuum had ≥2 nuclear accretions of irregularly shaped basophilic chromatin clumps, basophilic or eosinophilic cytoplasm, some cytoplasmic vacuolation, but seemingly intact cell membrane (Figure 1E,F) [30,53,64,66,68,83]. Apoptotic cells were identified as neurons because of their size and residual cytoplasm, or they were cell type non-identifiable and were round and small profiles with eosinophilic, condensed cytoplasm, chromatin clumps (≤4 crescent-shaped or round clumps), and cell surface that often was withdrawn from the surrounding neuropil [27,59].

### 2.8. Quantification of Proteinopathy in IHC Sections

A profile counting-based approach was used for quantification of the immunopositive profiles in human and piglet brain regions. For each antibody, three to four sections were counted for each human or piglet brain. All counts were performed at 1000× magnification in random no overlapping microscopic fields with careful z-axis focusing. Different markers for proteinopathy were counted in a variety of different brain regions in the human and piglet. This strategy was performed to determine if the pathology was generalized throughout the brain and thus likely to be widespread or global and in locations affecting multiple functional domains that could reflect the complex neurological syndromes of surviving HIE infants. In human brain, aggregated αSyn positive cell bodies were counted in frontal cortex layers 2 and 3, and aggregated αSyn positive cell bodies with possible inclusions were counted in the substantia nigra. Nitrated-Syn positive cell bodies were counted in human striatum, entorhinal cortex (ERC), and hippocampus CA1. In piglet brain sections, aggregated αSyn positive boutons were counted in somatosensory cortex, nitrated-Syn positive neuronal cell bodies were counted in ERC. Misfolded/oxidized SOD1 positive neurons were counted in CA1 and cerebellar cortex. PrP-positive Purkinje cells were counted in piglets. It was difficult to match regionally the neuropathology and proteinopathy in human and piglet brains because of tissue sampling design differences and tissue section variances.

### 2.9. Statistical Analysis

The data were analyzed using GraphPad Prism 9.5.1 or XLSTAT 2023.1.5 software. Data normality assessments were performed using the Shapiro–Wilk test. There was no exclusion of data points for any data set. Cells counts were analyzed by one-way ANOVA and post hoc Holm–Sidak test. When considering group side determinations, in prior neuropathology work on piglets [18,24] the mean difference in ischemic necrotic neurons within neocortex between HI-NT and sham-NT piglets was 100 with a within-group standard deviation of 5. A sample size of four piglets generates power >0.9. We increased the sample size to allow for some variability in our estimates. Western blot immunoreactive densities were normalized to Ponceau S staining and group comparisons were performed by Kruskal–Wallis ANOVA followed by post hoc Dunn’s testing. *p* values < 0.05 were deemed statistically significant.

## 3. Results

### 3.1. HT Shifts Neuronal Cell Death Pattern in Human HIE

We evaluated by H&E staining the neuropathology in infant HIE brains from autopsy cohorts that did not have HT treatment or did receive the standard-of-care HT treatment. Hippocampus, striatum, and neocortex were assessed. For relative normal histology reference, the HIE cases were compared to autopsy brains of infants with non-HIE causes of death, depending on brain regional sample availability. The non-HIE infant brain sections were examined thoroughly for neuropathology to determine their suitability as “controls”. The non-HIE infant “controls” cases with the most severe neuropathology in the brain only were the SMA cases (Appendix A). In the motor cortex of SMA cases, subsets of layer 5 neurons were chromatolytic (Appendix A), but the neocortex in general, as well as the hippocampus, putamen, and cerebellar cortex, were free of classic HI eosinophilic neurons.

In the CA1 region of hippocampus in infant non-HIE cases (Figure 1A,B), basophilic pyramidal neurons prevailed without morphologic evidence of eosinophilic HI injury. These neurons had an intact non-vacuolated cytoplasm and non-condensed nucleus (Figure 1B). The neuropil was homogenous, appearing uncorrupted as uniformly pink, smooth, and non-vacuolated in H&E-stained brain sections (Figure 1B). Perineuronal spaces were interpreted as postmortem artifact (Figure 1B, Appendix A). In contrast, the CA1 of neonatal HIE cases without HT was obliterated (Figure 1C,D). The neuropil was pale, vacuolated, and overall rarified (Figure 1C,D) compared to the integrity of the neuropil in non-HIE cases (Figure 1B). Many (~70%) pyramidal neurons in the CA1 of HIE-NT cases were necrotic, while few (<5%) degenerating neurons were apoptotic or hybrid forms of cell death described previously as continuum cell death (Figure 1D,G) [18,27,30,66,83]. Specifically, the number of necrotic CA1 pyramidal neurons in HIE-NT cases was significantly higher (*p* < 0.0001) than the number of non-necrotic degenerating neurons (Figure 1G). The neuropathology in the CA1 of HIE-HT cases differed fundamentally from the picture seen in HIE-NT cases. First, the neuropil was preserved relatively in HIE-HT cases (Figure 1E) compared to the rarefaction of HIE-NT cases. Second, the predominant form of pyramidal neuron degeneration was apoptotic or continuum-hybrid forms of cell death, rather than necrotic cell death (Figure 1F,G). The number of necrotic CA1 pyramidal neurons in HIE-HT cases was significantly lower (*p* = 0.001) than the number in HIE-NT cases (Figure 1G), but the number of non-necrotic degenerating cells was significantly increased (*p* < 0.001, Figure 1G) compared to the number of necrotic neurons in HIE-HT cases. The number of apoptotic-continuum degenerating neurons in the CA1 of HIE-HT cases was significantly higher (*p* < 0.0001) than the number in HIE-NT cases (Figure 1G). However, the shift in neuronal cell death form occurred in CA1 but not in the CA4 (the intradentate polymorphic layer) because the neurodegeneration was largely necrotic in HIE-NT and NIE-HT cases (Figure 1D,H).

In the caudate nucleus and putamen of neonatal non-HIE autopsy cases (Figure 2, Appendix A), the neuropil was homogenously smooth, non-vacuolated, and pink in H&E staining; in that parenchymal matrix were embedded principal neurons with medium-sized round or ellipsoid somas (Figure 2A, Appendix A) consistent with elegant descriptions of human striatum [84]. Occasionally, magnocellular neurons, likely the less numerous striatal cholinergic interneurons [84], were seen (Figure 2D). In the caudate nucleus and putamen of HIE-NT cases, the neuropil was pale and severely vacuolated; about 40–50% of the neurons showed classic ischemic-necrotic degeneration, while significantly fewer neurons were of the apoptotic-continuum form in the caudate nucleus (*p* = 0.0001) and putamen (*p* < 0.0001) (Figure 2B,F,G). The necrotic neurons had eosinophilic and shrunken/sharply angular cell bodies with a condensing nucleus. Some large magnocellular neurons evinced classic chromatolysis (Figure 2E) typical of axon-target disconnection-denervation [59,85], consistent with loss of principal medium-sized neurons in striatum (Figure 2F,G). The caudate nucleus and putamen in HIE-HT cases were distinguished by a partially preserved neuropil and a clear shift from necrotic cell death to apoptotic-continuum cell death (Figure 2C,F,G). The number of apoptotic-continuum degenerating neurons in HIE-HT cases was significantly higher than the number in HIE-NT cases in both the caudate nucleus (*p* = 0.0008) and putamen (*p* < 0.0001) (Figure 2F,G).

The neocortical pathology in infant HIE-NT and HIE-HT cases was generally similar, but HIE-NT and HIE-HT cases were very different from non-HIE cases (Appendix A). In non-HIE cases, the neocortex had divisible layers (Appendix A), and the pyramidal neurons were basophilic with a maintained triangular morphology (Appendix A). In HIE cases, the neocortex had prominent edema (Appendix A), and the lamination was often obfuscated compared to non-HIE neocortex. The cerebral edema was particularly evidenced by comparing the thickness of layer 1 in non-HIE and HIE-NT cases (Appendix A). Moreover, in HIE cases, there was ubiquitous loss of cortical pyramidal neuron morphology (Appendix A) and many neuron cell bodies were round and swollen with cytoplasmic pallor or disintegration. This neocortical picture in HIE cases, in general, was saliently distinct from the neocortex in the non-HIE infant cases we examined (Appendix A). In human HIE cases it was often difficult to discern swollen neurons from swollen glia, but immunofluorescence for NeuN was helpful in identifying degenerating cells as neurons (Figure 3E). Nuclear swelling, chromatin condensation, and NeuN staining patterns were also shown to be helpful in detecting neocortical neuron injury in HI neonatal piglets with and without HT treatment [18,68,82]. The residual cytoplasm of some cells in the human HIE neocortex, distinguishable as neurons, was eosinophilic (Appendix A). The chromatin condensation pattern of the nucleus of the majority of degenerating neocortical neurons in HIE-HT cases was noteworthy for its reticulation and irregular clumping (Appendix A) that was distinct from the chromatin condensation pattern seen typically in apoptotic cortical neurons (Appendix A).

### 3.2. HT Robustly Protects Cortical and Subcortical Neurons in Neonatal Piglet HI

The different piglet experimental protocols and their group sizes are summarized in Table 2. We evaluated by H&E staining the neuropathology in HI piglets treated with HT or NT during recovery as described [18,24,64]. Motor cortex, identified previously by cytoarchitecture, connectivity and electrophysiology [18], and the putamen were assessed for cytopathology and cell counting (Appendix A). Unfortunately, we could not identify specifically the motor cortex regions of the frontal cortex in the human HIE cases due to the autopsy sampling strategy, so the region was broadly identified as frontal cortex without the resolution of the motor cortex as part of the frontal cortex. In the motor cortex of HI-NT piglets, the neurodegeneration was uniformly necrotic; though variable in severity, there was highly significance damage in anterior motor cortex (*p* = 0.003) and posterior motor cortex (*p* = 0.004) compared to sham-NT piglets (Appendix A). In the motor cortex of HI-HT piglets, the ischemic–necrotic neurodegeneration was completely (anterior) or nearly completely (posterior) blocked (Appendix A). Yet, at the same time, no changes were detected in apoptotic profiles in the motor cortex of HI-HT piglets (Appendix A). In the putamen of HI-NT piglets, the neurodegeneration was severe (*p* < 0.0001), compared to the sham-NT piglets, and was uniformly necrotic (Appendix A). HI-HT piglets had significant (*p* = 0.0001) neuroprotection compared to HI-NT piglets (Appendix A). No corresponding increase in putamen apoptosis was observed in HI-HT piglets (Appendix A), consistent with the persistent neuroprotection that we have reported previously [17].

### 3.3. Synucleinopathy Occurs Rapidly in Excitotoxically Injured Human Oligodendrocytes and Neurons in Cell Culture

Clinical perinatal brain injuries are believed to be caused by excitotoxicity, oxidative/nitrative stress, and mitochondrial pathobiology [26,86]. Experimental acute neonatal brain injury is indeed driven in part by excitotoxicity, oxidative/nitrative stress, mitochondrial pathobiology, and proteinopathy [32,54,87]. These conditions are permissive for synucleinopathy in human PD and MSA [58,88]; therefore, we tested experimentally the hypothesis that synucleinopathy and SOD1 pathology occur in a cell autonomous reductionist version of excitotoxically injured human oligodendrocyte and neurons in essentially pure cell culture. Positive results in this paradigm establish an evidence-based foundation for examining synucleinopathy and aberrant SOD1 in human HIE brains.

As a pertinent antecedent to immunohistochemical staining for α-Syn pathology in human neonatal HIE brain, we developed two novel culture models of human neural cell excitotoxicity using iPS cells and embryonic stem cells. We used directed differentiation of iPS cells to an oligodendrocyte lineage (Figure 3). This cell system has not had the requisite characterization, so data are provided here. We verified their differentiation by differential interference microscopy of live cells (Figure 3A–C) and subsequent immunophenotyping of fixed cells. The cells passed through an early oligoprogenitor stage, identified by strong PDGFR-α immunoreactivity (Figure 3D). A later oligoprogenitor stage was identified by CNPase and Olig2 (Figure 3E) and O4 immunostaining. Then, a mature oligodendrocyte stage was identified by strong Bin1 immunoreactivity (Figure 3F). The cultures had minimal or no staining of makers for neurons (class 3 β tubulin) and astrocytes (GFAP) staining.

We developed a model of human oligodendrocyte injury using excitotoxicity. NMDA receptor excitotoxicity was used because we found that it causes neurodegeneration in piglets similar to that caused by HI and produces MRI-detected signal intensity changes similar to those seen in human HIE [66]; furthermore, striatal neurons in HI piglets show evidence for excitotoxic NMDA receptor activation identified by antibodies to specific phosphorylated NMDA receptors that report channel activation [89].

At an early oligodendrocyte stage of differentiation, cell cultures were exposed to QA to induce NMDA receptor-mediated excitotoxicity or to vehicle. Then, after a 12 h exposure and 12 h washout period with fresh media, the cells were assessed for excitotoxic injury evidenced by cell attrition (shrinkage) and vacuolation (Figure 3G,H). QA exposure induced significant (*p* < 0.001) structural damage to human differentiating oligodendrocytes (Figure 3I).

Western blots of human immature oligodendrocyte cell lysates probed with monoclonal antibodies to α-Syn, in putative aggregated or oligomeric, forms showed increased levels of immunoreactive proteins with electrophoretic motilities like the α-Syn monomer at ~16 kDa, another at ~32 kDa, and another at ~64 kDa, the two latter consistent with oligomeric species of α-Syn (Figure 3J). A monoclonal antibody to nitrated-α/β-Syn (Table 3) detected in vehicle-treated human oligodendrocytes doublet bands at ~50 kDa and another band at ~100 kDa, but no nitrated-Syn monomer was detected (Figure 3K). In contrast, in QA-treated human immature oligodendrocytes, an immunoreactive band was seen at ~20 kDa, and several additional higher molecular weight species were seen that were not detected in control cells (Figure 3K).

Similar excitotoxicity experiments were performed on human neurons directly differentiated from embryonic stem cell-derived neuroprogenitors in culture (Appendix A). Compared to vehicle-treated human neurons, QA induced significant cytoplasmic vacuolation (*p* < 0.0001) and apoptosis (*p* = 0.002) (Appendix A). In addition, strong immunoreactivities were detected with antibodies to aggregated α-Syn and nitrated-Syn (Appendix A) in human neurons after QA treatment. However, the molecular weights of the detected proteins were different from those seen in the QA-injured oligodendrocytes (Figure 3K,J). In QA-injured human neurons, a major α-Syn band was detected with an antibody to aggregated α-Syn at about 35 kDa and a lesser band at about 60 kDa. Nitrated-Syn in QA-injured human neurons was detected as a monomer and as higher molecular weight bands above 50 kDa. These findings encouraged our immunohistochemical and immunofluorescence experiments by showing the likely specificities of antibodies to TCPs in human neural tissues and the likelihood of detecting specific changes in brain injury involving excitotoxic mechanisms such as neonatal HIE.

### 3.4. Synucleinopathy Occurs in Human Neonatal HIE Brain

Brain sections of human neonatal HIE and non-HIE cases were immunostained with antibodies to aggregated α-Syn. In the frontal cortex of HIE cases, cells positive for aggregated α-Syn were significantly (*p* < 0.001) more numerous (Figure 4A,D) compared to the relative paucity of immunoreactivity in non-HIE cerebral cortex (Figure 4B,D). Most of the aggregated α-Syn-positive cells were degenerating cells, as discerned by nuclei with chromatin condensation suggesting apoptosis or variants of apoptosis (Figure 4C). Immunofluorescent labeling for NeuN demonstrated that many of the cortical cells positive for aggregated α-Syn were neurons (Figure 4E).

In the midbrain of human HIE cases, large neurons in the substantia nigra showed cytoplasmic eosinophilic pathology, including round eosinophilic structures of unknown identity (Figure 5A), and nuclear chromatin condensation reminiscent of variants of continuum cell death (Figure 5A,B) described previously in PD [28,31,66]. Nevertheless, some nearby neurons appeared relatively normal (Figure 5C). Large neurons in the substantia nigra of HIE cases had aggregated α-Syn immunoreactivity; their number was increased significantly (*p* = 0.0002) compared to non-HIE cases (Figure 5E). Some cells in the substantia nigra possessed large cytoplasmic accumulations positive for aggregated α-Syn (Figure 5D). These cytoplasmic perinuclear accumulations were also positive for ubiquitin (Figure 5D, inset). Dual labeling for the dopaminergic neuron marker TH and aggregated α-Syn identified some morphologically normal appearing neurons with focal dendritic domains positive for aggregated α-Syn in HIE cases (Figure 5F). Immunofluorescence co-labeling confirmed the presence aggregated α-Syn immunoreactivity in TH-positive neurons in the substantia nigra in HIE cases (Figure 5G). Immunofluorescence co-labeling also demonstrated the presence of nitrated-Syn in focal regions of dendrites and cell bodies in apparent attritional neurofilament-positive neurons in substantia nigra (Figure 5H). Using serial 5-µm thick paraffin sections of the human midbrain, in which the same large dopaminergic nigral neurons can be represented in different sections, we found a suggestion that the aggregated α-Syn- and nitrated-Syn-positive structures were often the same.

Immunostaining for nitrated-Syn also revealed robust synucleinopathy in neonatal human HIE forebrains (Figure 6). Significant increases were seen in the number of nitrated-Syn-positive cells in HIE cases compared to non-HIE cases in the striatum (Figure 6E, *p* = 0.0007), entorhinal cortex (Figure 6F, *p* = 0.0003), and hippocampal CA1 (Figure 6G, *p* = 0.0005). In the entorhinal cortex and CA1, the nitrated-Syn immunoreactive cells had nuclei displaying overt chromatin condensation indicative of degeneration, but in the putamen, nitrated-Syn-positive cells did not have readily apparent chromatin condensation signatures, but they had cytoplasmic aggregations (Figure 6A). Negative control sections (HIE brain sections that were incubated with non-immune mouse IgG instead of primary antibody) were blank (Figure 6B). Dual immunohistochemical labeling was performed to identify the specific types of cells harboring nitrated-Syn in human HIE cases. DARP32, a marker for medium-sized striatal neurons, colocalized with nitrated-Syn in the putamen (Figure 6H). Immunofluorescent colocalization identified nitrated-Syn in neurofilament neurons in the entorhinal cortex (Figure 6I).

### 3.5. Putative Toxic Forms of SOD1 Accumulate in Human Neonatal HIE Brain

Human oligoprogenitors in cell culture were injured excitotoxically by QA or exposed to the vehicle, lysed after 24 h, and immunoblotted for oxidized/misfolded SOD1 using C4F6 and B8H10 monoclonal antibodies (Table 3, Figure 3L). Both C4F6 and B8H10 antibodies gave similar patterns, though C4F6 gave stronger immunoreactivity. Oxidized/misfolded SOD1 was not detected in vehicle-treated human oligoprogenitors but striking immunoreactivity was seen in QA-treated cells (Figure 3L). A major band was seen between 15 and 20 kDa, consistent with a post-translationally modified SOD1 monomer [90], and higher molecular apparent oligomers were seen spanning 20–50 kDa (Figure 3L) [90]. A very large oligomer of SOD1 immunoreactivity was seen at ~250 kDa (Figure 3L); this form of SOD1 has also been described before in mouse ALS [90].

Human brain sections of neonatal HIE and non-HIE cases were immunostained for oxidized/misfolded SOD1 (Figure 7). The regions assessed were the neocortex (Figure 7A–D), caudate nucleus (Figure 7E,F), hippocampus (Figure 7G), and cerebellum (Figure 7H–L). In the cerebral cortex of HIE cases, many cells were strongly positive for oxidized/misfolded SOD1 (Figure 7A–C), but non-HIE cases were generally negative. Oxidized/misfolded SOD1 immunoreactivity was notably present in the nucleus of cells that were positive for the T-box transcription factor TBR1 and, thus, are cortical neurons (Figure 7D). Some condensed degenerating cortical neurons had large round clumps of oxidized/misfolded SOD1 immunoreactivity in the nucleus (Figure 7C). This finding in the human brain corroborates our cell culture result showing that human neurons exposed to QA accumulate oxidized/misfolded SOD1 (Figure 3E) and undergo apoptosis (Appendix A). Other cells in the cerebral cortex that were positive for oxidized/misfolded SOD1 but not for TBR1 (Figure 7D) suggested aberrant SOD1 was localized to glia. Dual labeling IHC suggested that astrocytes in the cerebral cortex, identified by the glutamate transporter GLAST, were positive for oxidized/misfolded SOD1 (Figure 7E). Immunofluorescence confirmed that GLAST-positive astrocytes were positive for oxidized/misfolded SOD1 (Figure 7F) and that the presence of oxidized/misfolded SOD1 was focally localized within astrocytes (Figure 7F). Immunofluorescence also demonstrated that Olig2-positive oligodendrocytes in cerebral cortex were also positive for oxidized/misfolded SOD1 (Figure 7G). Selective neuronal vulnerability was also seen in subcortical forebrain regions such as the caudate nucleus where many medium-sized striatal neurons were positive for oxidized/misfolded SOD1, but nearby neurons were not positive (Figure 7H). Subcortical white matter pathways in HIE cases showed salient vulnerability to proteinopathy; Olig2-positive oligodendrocytes were positive for oxidized/misfolded SOD1 (Appendix A). Diffuse plaque-like structures positive for oxidized/misfolded SOD1 were seen throughout the white matter (Appendix A).

The hippocampus and cerebellar cortex of human HIE cases provided stunning information (Figure 7). CA1 neurons were strongly positive for oxidized/misfolded SOD1; some neurons showed distinctive intracellular inclusion-like structures that were decorated with oxidized/misfolded SOD1 (Figure 7I, insets). These putative inclusions had crystal-like shapes, including block (Figure 6G, upper right inset) and circle forms (Figure 7I, lower left inset), consistent with the structure of TCPs [47]. Examination of the cerebellar cortex yielded additional novel aspects of proteinopathy in human HIE cases (Figure 7L–Q). Some Purkinje cells, identified by NF68, were strongly positive for oxidized/misfolded SOD1 with round cytoplasmic or nuclear accumulations of immunoreactivity (Figure 7J,K). Other Purkinje cells were more lightly positive (Figure 7L) or had focal dendritic accumulation of oxidized/misfolded SOD1 as seen by the colocalization with NF68 (Figure 7M). Purkinje cells with a classic ischemic–necrotic morphology were completely negative for oxidized/misfolded SOD1 (Figure 7N). Many oxidized/misfolded SOD1 neuritic abnormalities were seen in the cerebellar cortex mirroring known innervation patterns [18] such as climbing fibers and mossy fibers (Figure 7O) and parallel fibers (Figure 7P). Their morphology and the colocalization of oxidized/misfolded SOD1 with NF68 indicated that these neuritic structures were axons (Figure 7Q).

### 3.6. Synucleinopathy in Brain Occurs Acutely in Piglet HIE and Is Presynaptically Localized

Synucleinopathy was found in three different HI piglet brain injury models. We used a translationally relevant brain HI model with standard-of-care treatment in piglets with 29 h survival and treatments of HI + NT, HI + HT, sham + NT, or sham + HT to determine if synucleinopathy forms rapidly and is affected by body temperature. By Western blot, aggregated oligomeric α-Syn was detected in piglet forebrain tissue lysates at ~65 kDa (Figure 8A). Aggregated α-Syn level was greatest in HI-NT piglet brain (Figure 8B). Formation of aggregated α-Syn was significantly (*p* = 0.0007) lower with HI-HT treatment (Figure 8B), but aggregated α-Syn accumulation with HI-HT treatment remained significantly higher (*p* = 0.0002) than sham-HT treatment (Figure 8B). Aggregated α-Syn accumulation after neonatal HI was seen again in a different model without HT and a survival of 4 days (Figure 8C). In this model, the piglets are recovered and extubated in about 3 h, with normal O_2_ saturations (no persisting hypoxia), and piglets are usually ambulatory, feed independently, and gain weight during the 4-day survival with uncommon indication of malaise, clinical seizures, or hyperthermic co-morbidity [62]; thus, they are phenotypically normal. Significant (*p* < 0.0001) accumulation of proteins immunoreactive to antibodies detecting aggregated α-Syn was found in somatosensory cortex 4 days after HI compared to sham procedure (Figure 8D). In our HI-QA model, aggregated α-Syn immunoreactivity was found to be significantly elevated 14 days after injury in degenerating neurons and in terminal-like boutons (*p* = 0.001) in the striatum (Figure 8E,G), while immunoreactivity was scarcely detected in sham piglets (Figure 8F,G). Immunofluorescence showed colocalization of aggregated α-Syn immunoreactivity with SV2 in cerebral cortex (Figure 8H) and synaptophysin in striatum (Figure 8I) demonstrating that these neuropil structures were presynaptic terminals. Immunoreactivities for other presynaptic proteins, including Munc18, cysteine-string protein, and β-synuclein, also colocalized with aggregated α-Syn immunoreactivity in piglet brain. In the human HIE brain, immunofluorescence also revealed colocalization of aggregated α-Syn immunoreactivity with SV2 in the cerebral cortex (Figure 8J), demonstrating that our observations in the piglet brain are relevant to human HIE.

Accumulation of nitrated-Syn was also detected prominently in the piglet HI by Western blot using the monoclonal antibody Syn12 (Figure 9A). In the naïve piglet forebrain, low levels of immunoreactivity for nitrated-Syn were detected with faint bands at about 40 and 50 kDa (Figure 9A). In piglets treated with sham-NT, overall nitrated-Syn was also low, but an additional doublet band of immunoreactivity was seen at about 30 kDa (Figure 9A). Sham-HT piglets had significantly greater nitrated-Syn in forebrain compared to sham-NT in the 30 kDa (*p* = 0.008) and 50 kDa (*p* = 0.01) size ranges (Figure 9B,C). HI and NT recovery in piglets induced a dramatic increase in the levels of nitrated-Syn. Syn nitration was also high in HI-HT piglets but levels were significantly lower than in HI-NT piglets (Figure 9B,C).

### 3.7. Putative Toxic Forms of SOD1 Accumulate in Piglet HIE Brain

Misfolded/oxidized SOD1 was detected in the HI piglet forebrain by Western blot using monoclonal antibody C4F6 (Figure 10A). Monomeric (~16 kDa) and putative oligomeric (40–250 kDa) forms were seen. The intensities of immunoreactivity among the piglet brains for the putative oligomers of different sizes were variable, suggesting that these bands were unlikely to be nonspecific proteins. Total levels of presumed oligomers (assessed collectively by densitometry) were not significantly different among the groups (Figure 10B). In contrast, the oxidized/misfolded SOD1 monomer was increased significantly in the HI-NT piglet forebrain (Figure 10C), and this increase was attenuated significantly in HI-HT piglets (Figure 10C). In HI piglets surviving for 4 days (without any therapeutic intervention), misfolded/oxidized SOD1 oligomers and monomers were increased significantly (*p* = 0.001) and (*p* = 0.009), respectively, compared to sham (Figure 10D–F).

Motivated by the immunohistochemical findings with misfolded/oxidized SOD1 in human HIE cases (Figure 7), we examined the localizations of SOD1 proteinopathy in piglet encephalopathy. We used our HI-QA model because the piglets had the longest survivals of 14 days (Table 2, Figure 11). Remarkably, as in human HIE (Figure 7G), subsets of pyramidal neurons in HI-QA piglets were positive for misfolded/oxidized SOD1 (Figure 11A), while age- and time-matched sham piglets had scant immunoreactivity (Figure 11B,E). Cell counting evinced a significant (*p* = 0.002) accumulation in the number of misfolded/oxidized SOD1-positve CA1 neurons in HI-QA piglets compared to sham piglets (Figure 11E). In the HI-QA piglet CA1 pyramidal neurons, the nuclear matrix, nucleolus, and cytoplasm were positive for misfolded/oxidized SOD1 (Figure 11A). Interestingly, in some pyramidal neuron nuclei, the misfolded/oxidized SOD1 was confined to the nucleolus without labeling in the nuclear matrix (Figure 11A). In the cerebellar cortex, HI-QA piglets had salient misfolded/oxidized SOD1 immunoreactivity in the nucleus of Purkinje cells (Figure 11C,F), while the sham piglets did not (Figure 11D,F). Cell counting demonstrated a significant (*p* < 0.001) increase in the number of misfolded/oxidized SOD1-positve Purkinje cells in HI-QA piglets compared to sham piglets (Figure 11F). The misfolded/oxidized SOD1 immunoreactivity in the nucleus of Purkinje cells formed speckle and granular inclusions (Figure 11C).

### 3.8. PrPopathy, Including Accumulation Proteinase K-Resistant PrP Immunoreactivity, Occurs in Neonatal Piglet Encephalopathies

We studied whether the iconic TCP, PrP, shows proteinopathy in two different models of piglet encephalopathy. Western blotting for PrP in the somatosensory cortex of naïve and HI-QA injured and sham piglets with 48 h survival showed a mature PrP band at ~37 kDa, and lower broad bands (~25–35 kDa), likely representing immature non- and mono-glycosylation forms of PrP (Figure 12A) [80]. In the naïve and sham piglet neocortex, these bands were the only immunoreactivities seen. In contrast, in HI-QA piglet neocortex, higher molecular weight forms of PrP were detected, likely corresponding to SDS-resistant oligomers of PrP (Figure 12A) [79].

The cerebellum is a region that shows exquisite vulnerability in PrP-related spongiform encephalopathies in humans and sheep [91]. In our HI piglet model, Purkinje cells show selective vulnerability with robust eosinophilic degeneration (Figure 12L). We, therefore, studied the cerebellum for PrPopathy in neonatal piglets after HI and 4 days survival. Subsets of Purkinje cell bodies became enriched in PrP in HI piglets (Figure 12B); in contrast, Purkinje cell bodies in sham piglets were generally either PrP-negative or had only faint immunoreactivity (Figure 12E,F), consistent with the finding that the PrP is normally transported anterogradely within the axon rapidly [92,93]. These PrP-enriched Purkinje cells in HI piglets had degenerating morphologies, including cytoplasmic vacuolation (Figure 12C) and cell body attrition and detachment from the neuropil (Figure 12D), consistent with their eosinophilic degeneration (Figure 12L). The number of PrP-positive Purkinje cell bodies was significantly higher (*p* = 0.00008) compared to sham piglets (Figure 12G).

Purkinje cells project to the deep cerebellar nuclei [94,95]. We examined these regions in our PrP preparations of HI piglet cerebellum and found axonal and bouton dystrophy (Figure 12H). Specifically, swollen peridendritic boutons were evident (Figure 12H). Intriguingly, plaque-like pathology was also seen in the deep cerebellar nuclei and surrounding subcortical white matter in HI piglets (Figure 12I). Like plaques in human and sheep PrP encephalopathies, respectively [91], these PrP-positive plaques were neuritic (Figure 12J) or diffuse (Figure 12K).

To further explore the significance of the PrP accumulation in Purkinje cells after HI in piglets, we tested its sensitivity to proteinase K digestion. PrP immunoreactivity in the molecular and granule cell layers and in the superficial subcortical white matter were nearly completely digested by proteinase K (Figure 12M,N). However, in the same sections, subsets of Purkinje cells and their dendrites were strongly resistant to proteinase K digestion (Figure 12N,O), demonstrating that the PrP in degenerating Purkinje cells is a more stable misfolded form compared to the native PrP in other parts of the cerebellum.

## 4. Discussion

This study was a cytopathological and molecular neuropathology analysis of human neonatal HIE and related neonatal piglet models of encephalopathy. The cytopathology was performed to determine if therapeutic HT alters the neurodegeneration phenotype in neonatal HIE. The molecular pathology focus was on proteinopathy, defined as pathological alterations to proteins that could convey a toxic gain or loss of molecular function, in clinical neonatal HIE and its piglet and cell models. A focus was synucleinopathy as seen by antibody-based detection of α-Syn protein as nitrated or stably aggregated in a form other than its monomeric state; both are thought to convey pathological abnormalities [55,56,57,58]. Our findings are as follows: (1) the neurodegeneration phenotype is altered by HT in human infant HIE but not in a piglet model of neonatal HI; (2) forms of proteinopathy, those usually reserved for adult-onset neurodegenerative disease, are florid and global in neonatal human HIE and its piglet and cell models; (3) proteinopathy signatures reminiscent of adult-onset neurodegenerative disease can form rapidly on an acute/subacute time scale; and (4) proteinopathy in neonatal HIE has a broad cell type distribution (neurons and glia) as shown by immunofluorescence. These findings could have important bearings on the neural system vulnerability, connectome spreading, and persistent, possibly evolving, damage in neonatal HIE with lifelong childhood to adult consequences even after HT treatment.

### 4.1. Experimental Design Considerations

A key aim was to compare the neuropathology in infants that did not receive HT, these were both archival cases of infants prior to HT becoming the standard of care and those individuals that missed the window of time for HT administration, to HIE infants that did receive HT and subsequently died. The human HIE neuropathology was compared to that seen in neonatal piglet models of encephalopathy caused by HI and excitotoxicity. We used a variety of different piglet models of neonatal encephalopathy to obtain a broadly encompassing insight into the progression and types of neurodegeneration to ensure that degenerative phenotypes were not missed and that proteinopathy could be identified biochemically and histologically in different precisely controlled experimental settings (Table 2), unlike the stochastic nature of clinical HIE. Piglets with HI were treated with overnight HT, or they had NT recovery [18,32,54,64]. Survival times for piglets differed, again, to glean a comprehensive purview of neuropathology, including proteinopathy, and because the survival times of human autopsy cases are random and not predetermined. The QA excitotoxicity injury was used in combination with HI (or without) in piglets for several reasons (Table 2). First, it allowed for a much less severe asphyxic cardiac arrest insult with improved recovery so that the piglets could have longer-term survivals, allowing for delayed neurodegeneration and potential proteinopathy, and without weakened cardiac state. Second, evidence suggests that NMDA receptor excitotoxicity is a mechanism for experimental neonatal HI brain injury [25,26,29,96,97], so focal intracranial QA lesioning was a complementary add-on, rather than an incongruent second injury. Third, our human oligodendrocyte and neuron cell culture experiments with essentially pure cells allowed the opportunity to explore directly if excitotoxic NMDA receptor activation can potentially cause synucleinopathy and oxidation/misfolding of SOD1 autonomously in these cells, thus providing proof-of-concept and new human cell models relevant to neonatal brain injury where oligodendrocytes and neurons show vulnerability [26]. Moreover, our cell experiments demonstrated that aberrant α-Syn and SOD1 can accumulate rapidly, within 24 h, and independent of hypoxia because the cells were not exposed to hypoxia. Cell culture work by others [98] has revealed that QA can facilitate the formation of nanostructures that participate in intercellular transfer of cytotoxic α-Syn oligomers, and, importantly, here we have seen α-Syn oligomers in human and piglet encephalopathies.

### 4.2. HT Shifts Neurodegeneration Type in Human HIE

We discovered that HT treatment of human neonatal HIE cases appears to fundamentally change the cytopathology of neurodegeneration. This alteration appears as a shift in the neurodegeneration in some brain regions away from classic ischemic necrosis, seen characteristically in HIE cases not treated with HT, to a structural hybrid of necrosis and apoptosis that we designated previously as continuum cell death [27,28,66,82]. This cytopathology shift appeared prominently in hippocampus CA1 and striatum. Cell death shifting did not appear to occur in the hippocampus CA4 sector. The predominant ischemic necrotic neurodegeneration seen in non-cooled infants is consistent with descriptions of human HIE neuropathology long before mild HT became the standard of care for moderate to severe HIE [99,100,101] and with more recent descriptions [102,103]. Cell death type shifting is consistent with the concept of the cell death continuum [26,52,83,104] as driven mechanistically by variations in the strengths and tempos of excitotoxicity, reactive oxygen species (ROS) production, intracellular Ca^2+^, cell volume control, and mitochondrial dysfunction and permeability transition pore activation [27,28,31]. Precedent for cell death shifting has been seen in a variety of nonneuronal cells [103,104,105]. Prior instances of cell death shifting in neurons (apoptosis to necrosis and necrosis to autophagy) has been seen in rodent models of neurodegeneration where oxidative stress and mitochondriopathy are involved [60,106] and in neonatal rat and adult mouse where excitotoxicity and different subtypes of glutamate receptors are activated [53,77]. The clinical cooling of infants with HIE might change the acute and subacute landscape of NMDA receptor excitotoxicity, oxidative stress from ROS, and proteinopathy as shown in HI piglets with cooling [97]. The difference in human versus piglet responsiveness to HT might be related to injury onset and tempo and the completeness and depth of HT cooling resulting in different efficacies.

### 4.3. HT Does Not Shift Neurodegeneration Type in Piglet HI Models

HT as a therapeutic in piglet HI is very effective in neuroprotection. The neuroprotection seen here was in the motor cortex and putamen. This finding is consistent with work by us [17,18,64] and others [13,15] using newborn piglet HI models. The protective effectiveness of mild whole-body HT (cooling for 38 °C to 34 °C) in gray matter of HI piglets seems nearly universal because most piglets treated with HT, even with a delay of 2 h and shorter durations than most clinical use, show benefit with essentially a number needed to treat (NNT) for effect as one [17,18]. This effectiveness is unlike human term HIE where the NNT for HT efficacy is ~7 to 14 infants depending on the outcome metric [8,11,12,107]; moreover, 40–50% of infants treated with HT die and survivors still have neurologic disability [12]. There are many possible explanations for the differences in HT efficacy among human HIE and piglet HI, including (1) undocumented start time and duration of insult in human HIE versus known start and duration for piglet HI; (2) incomplete knowledge of physiology at baseline and injury emergence in clinical HIE versus known physiological baseline and tracking in experimental HI; (3) unclear outbred genetic background in human HIE versus better established more uniform genetic backgrounds for piglet HI; (4) use of anesthesia during HT for piglet HI but not in clinical settings and other protocol differences such as rate of rewarming; (5) prolonged agonal state with postmortem delay and brain immersion fixation at autopsy in human HIE versus optimally prepared piglet brains; and (6) as shown here, the apparent unmitigated formation TCP proteinopathy in human HIE, despite HT, while proteinopathy appears diminished in piglets with HT.

The difference seen with HT in human term HIE, but not neonatal piglet HI, regarding apparent cell death shifting has a specific pattern. In non-cooled infants, the predominant phenotype of the neurodegeneration was cell necrosis, consistent with the traditional [99,100,101,108] and more recent [102,103] neuropathological descriptions. Necrotic forms of neurodegeneration indeed have nuclear pyknosis, chromatin condensation, and karyorrhexis, but the patterns are distinct from classic apoptosis as described in nervous system development [27,52,109,110,111,112]. In contrast, in cooled infants, the predominant phenotype of the neurodegeneration in the forebrain appeared morphologically as syncretic forms with characteristics of necrosis and apoptosis, but distinctly not independently either form of degeneration; thus, the degenerating neurons appeared as hybrids or continuum cell death in cooled human infants.

This type of syncretic cell death is robust in neonatal rat brain injured by excitotoxicity or HI [52,53] and in neonatal piglets with QA excitotoxicity [66]. The emergence of this neurodegenerative phenotype in cooled term HIE infants is interpreted as shifting neuronal cell death along the cell death continuum. Continuum cell death might not be seen readily in cooled HI piglets because of different NMDA and non-NMDA glutamate receptor and proteasome profiles [52,67,102] compared to human infant neurons, and because of the precise timing of implementation of the HT in relation to the known injury in piglets could interrupt effectively fundamental upstream sentinel pathological events leading to strong neuroprotection. There are similarities in pig and human neuron responses to injury, namely excitotoxicity and DNA damage, but there could also be differences in neurodegeneration, including caspase recruitment [19]. The possible sentinel events interrupted, particularly with early HT implementation in piglets, could be excessive NMDA receptor activation, ROS production, DNA single-strand break formation and repair, and proteinopathy [54,89,97].

### 4.4. Aberrant, Putatively Toxic, Proteins Can Accumulate Rapidly in Neonatal HIE and Piglet HI

Proteinopathy could be a new hub for acute and delayed selective vulnerability and network neurodegeneration in infant survivors of HIE and neonatal experimental models of HI for translational therapeutic targeting. White matter pathways interconnecting cortical–cortical and cortical–subcortical brain regions accumulate oxidatively damaged proteins (in the form of protein carbonyls) within 29 h after HI in neonatal piglets [54]. Elevated protein carbonyls are also detected in cerebral cortex 24 h after cortical contusion [87] and HI [113] in neonatal rodents; moreover, αSyn oligomers accumulate between one and three months of age after neonatal cortical contusion injury in transgenic mice expressing human mutant αSyn [87]. This prior work was insightful conceptually by introducing the possibility of time contraction (or quickening) involving the formation of TCPs such as those causing synucleinopathy. Thus, synucleinopathy is a prominent feature of neonatal brain injury and occurs on a more rapid timescale (hours to days) than realized previously, in contrast to the years-long protracted timescale anticipated for human neurodegenerative disease. However, this finding in mice [87] was with supra-physiological expression levels of human mutant αSyn. On this foundation, we hypothesized that TCPs could be seen rapidly in the brain with physiological expression levels after neonatal HI.

To this end, we used well-characterized and highly specific antibodies to pathological TCP forms of αSyn and SOD1 (Table 3). The antibodies to aggregated-αSyn and nitrated-Syn have been used in studies of PD, MSA, and Lewy body disease [57,71,72]. The monoclonal antibodies to misfolded/oxidized SOD1 have been used in studies of ALS [33,75,114]. Here, we first used human oligodendrocyte and neuron cell systems with excitotoxic injury to further characterize the antibodies to pathological forms of αSyn and SOD1 and for proof-of-concept evidence. Importantly, because we did not have frozen brain tissue from our clinical neonatal HIE cohorts for Western blot, our cell culture systems allowed us to characterize using Western blot the antibodies in human neural cells exposed to a neonatal relevant injury. This strategy also provided important validation for the appropriate application of the antibodies for IHC and immunofluorescence. NMDA glutamate receptor excitotoxic activation with QA-induced accumulation of nitrated Syn and aggregated αSyn within ~24 h in oligodendrocytes and neurons. SDS-resistant high molecular weight oligomers were detected after QA. αSyn oligomers have been seen in other experimental settings [9,34,56,115]. For example, QA treatment of human neuroblastoma cells appears to enable the formation of aggregated αSyn seeds that could be cytotoxic and permissive to cell-to-cell spreading of injury [98]. We found that misfolded/oxidized SOD1 also accumulated, including oligomers, in QA exposed human oligodendrocytes. Our detection of misfolded/oxidized SOD1 oligomers with C4F6 and B8H10 monoclonal antibodies (Table 3) is consistent with other studies [90]. The results of our cell culture experiments were encouraging, so we advanced experiments to postmortem human HIE brains and piglet encephalopathy.

αSyn and SOD1 TCPs were detected and localized discretely by IHC and immunofluorescence in the infant human brain. Aggregated αSyn was seen in human HIE brains. Aggregated αSyn is thought to be toxic to neurons and trigger inflammation [116,117], though it is controversial how aggregated αSyn might cause neurodegeneration [55]. Nevertheless, we found striking aggregated αSyn immunoreactivity directly in degenerating human cortical neurons (Figure 5E) and in substantia nigra dopaminergic neurons (Figure 6F,G) in HIE infants. The aberrant αSyn could be precisely localized to cytoplasmic aggregates that colocalized with ubiquitin by immunofluorescences and to foci on dendritic domains. This has not been shown before. The localization of nitrated-Syn in human HIE brain was revealing because it was found accumulated in dopaminergic nigral neuron dendritic foci, striatal neurons, often forming cytoplasmic aggregations (Figure 6D), and because neurons degenerating with a continuum cell death phenotype in entorhinal cortex and hippocampus CA1 were strongly positive (Figure 7C,D,I). Nitrated-Syn implies intriguingly the presence of peroxynitrite as a toxic mechanism [58,88,118]. It has been known that nitrated-Syn forms pathological inclusions in human neurodegenerative diseases [56,58], but our results show for the first time in vivo the enrichment of nitrated-Syn in neurons with the distinctive cell death nuclear morphology of continuum degeneration in the neonatal human brain.

The misfolded/oxidized SOD1 localization in human HIE brain was distinguished by four prominent features: (1) it was strongly nuclear; (2) it formed discrete subcellular structures in the cytoplasm and nucleus; (3) it was localized to neurons, oligodendrocytes, and astrocytes as demonstrated by immunofluorescence; and (4) it appeared in dystrophic neurites. These four features of misfolded/oxidized SOD1 immunoreactivity have been seen also in human ALS tissues [33,119], ALS patient iPS cell-derived motor neurons [120], and mouse models of ALS [61].

We examined piglet HI brains for similar findings to rule-out the potential for human brain autopsy postmortem artifact, particularly with the immature brain, and to improve our understanding of the temporal pathological process. Aggregated and nitrated αSyn and SOD1 TCPs were detected by Western blot and IHC in the HI piglet brain. Our early-endpoint Western blot experiments in HI piglets revealed that αSyn and SOD1 TCPs could be formed in the forebrain within 20 h after injury. The immunohistochemical pattern seen in piglets with aggregated αSyn antibodies was similar to that seen in human HIE. Cortical neurons positive for aggregated αSyn were seen like those in humans; moreover, HI piglets and human HIE cases also had prominent presynaptic terminals containing aggregated αSyn immunoreactivity as demonstrated by immunofluorescent double labeling with SV2, synaptophysin, and other synaptic markers. αSyn is known to be a presynaptic protein and is highly enriched (1% of cytosolic protein) in brain and forms Lewy body inclusions [121,122,123]. The structural details of the aggregated αSyn synaptic labeling might be more labile in the postmortem human tissue but nicely preserved in our piglet models because the brains are prepared optimally. The neuronal cell body labeling for nitrated-Syn was similar among human HIE and piglet HI though there was also generally more neuropil labeling in piglet. The patterns of immunolabeling seen for misfolded/oxidized SOD1 in human HIE were mirrored also in piglet HI. These findings in neonatal human and piglet encephalopathies provide new insight into the rapidity at which TCPs can be formed in the brain and demonstrate that, at natural physiological expression levels of αSyn and SOD1, putative toxic forms can accumulate in response to clinically relevant neonatal pathophysiological insults. Thus, the piglet HI is a faithful animal model for human neonatal brain synucleinopathy and other proteinopathies.

It is also important to contextualize our findings with proteinopathy in control conditions. In our cell culture experiments, both oligodendrocyte and neuron, we detected some aggregated αSyn and nitrated in the vehicle conditions. This suggests that there is a constitutive low-level presence of proteionopathy that is likely due to endogenous oxidative stress (cells were grown in atmospheric oxygen concentration) and protein misfolding. In our piglet HI-HT model, we detected low levels of abnormal αSyn and SOD1 in sham piglets. It is noteworthy that these sham piglets had ~29 h of anesthesia and machine ventilation and some had overnight HT. Both physiological perturbations could produce abnormalities in proteostasis [54].

It is relevant to consider the modifiability of proteinopathy in the injured neonatal brain, particularly if it becomes an established driver of acute and delayed neurodegeneration. αSyn and SOD1 TCPs were detected in human HIE brains regardless of HT treatment. Proteinopathy appeared to be modifiable by HT in piglets. We found previously in HI piglets that HT mitigated protein carbonyl formation [54,97], so it is possible that oxidative damage is driving the proteinopathy.

### 4.5. Pathology in PrP, the Iconic TCP That Can Mediate Trans-Synaptic Spreading of Disease, Is Seen in Neonatal Brain HI

Because PrP is an iconic TCP in cases of clinical and experimental spongiform encephalopathy [46,47], we examined PrP in our piglet models of neonatal encephalopathy. By Western blot, monoclonal antibody F89/160.1.5 (Table 3) detected a broad band consistent with PrP and its non-, mono-, and di-glycosylated forms [80]. In the naïve and sham piglet brain, only these forms of PrP were detected. Remarkably, in HI-QA injured brains, higher molecular weight species of PrP were observed consistent with the formation of oligomeric TCPs [124,125]. The immunohistochemical pattern in normal piglet cerebellum was extraordinary for its enrichment in the neuropil and low immunoreactivity in neuronal cell bodies. Our observations in pig brain are consistent with that reported for hamster cerebellum [126]. The low neuronal cell body positivity and the high neuropil enrichment are consistent with the finding that PrP undergoes rapid anterograde axonal transport [92,93]. HI piglets, in contrast, had a spectacular phenotype in the cerebellum where Purkinje cell bodies were strongly positive for PrP and they generally showed degenerative phenotypes. We then did proteinase K digestion experiments. We found that, while much of the PrP immunoreactivity in the cerebellar cortex molecular and granule cell layers was proteinase K-sensitive, the PrP accumulating in degenerating Purkinje cell bodies and their dendrites was essentially proteinase K insensitive; thus, it possibly represents a toxic misfolded form of PrP. Accumulation of a TCP form of PrP in ischemic neurons has not been shown before and possibly represents a new mechanism of delayed ischemic neurodegeneration. Another feature, likely related to the Purkinje cell body pathology, was the visibility of apparent PrP-positive terminal degeneration in the deep cerebellar nuclei that are targets of Purkinje cell axons [94,127]. In human infant HIE, cerebellar Purkinje cells and the deep cerebellar nuclei, particularly the dentate nucleus, are often damaged [108]. Astonishingly, within the deep cerebellar nuclei and surrounding white matter in HI piglets, we observed neuritic plaque pathology like those seen in human prion spongiform encephalopathies [128]. The neuropil and neuronal/glia cell body spongiform pathology of neonatal encephalopathies is remarkable [18,66,68,82], perhaps proteinopathy is its cause.

### 4.6. Proteinopathy as a Possible Mediator of Neurodegeneration and Delayed Trans-Synaptic Network Selective Vulnerability in Neonatal HIE

This study supports the concept that proteinopathy could have roles in the acute, subacute (secondary), and delayed (tertiary) injury in the neonatal brain associated with HIE. In the acute phase (~3–6 h after HI), tubulin damage, including nitration, is seen in piglets [29]; this implies that brain cytoskeletal architecture is disrupted by peroxynitrite. This phenomenon is consistent with work showing that tubulin is prone to tyrosine nitration by peroxynitrite [118]. In addition, ROS introduce carbonyl accumulation subacutely (20–29 h after HI in piglets) in proteins like pericentriolar material-1 (PCM1) and triosephosphate isomerase-1 (TPI1) [32]. Damage to PCM1 might be significant to neuro- and glio-degeneration after HI because PCM1 organizes the cytoskeleton and is critical for dendrite stability and neurite elaboration [129,130], possibly driving the collapse of neuronal dendrites and destruction of oligodendrocyte processes and myelination after HI. TPI1 is interesting because it is prone to aggregation after oxidative damage and might seed tau aggregation [131]. We now show here that misfolded, nitrated, aggregated, and oligomerized proteins accumulate in brain at subacute (20–29 h) and delayed times (4–14 days) after neonatal HI. Aberrant α-Syn, SOD1, and PrP cause neurodegeneration and cognitive deterioration in humans linked to PD, MSA, ALS-frontotemporal dementia, and CJD and are recognized as targets for therapy. Many of the pathological proteins involved in these diseases and neonatal HI are IDPs and TCPs [48]. IDPs are proteins that lack, in part or fully, a fixed three-dimensional structure [132]. αSyn is an IDP [48]. TCPs are proteins that can adopt a pathological form, such as a fold, turn, bend, barrel, or β-pleated sheet conformation under physiological conditions that then become toxic to cells [133].

TCPs, including PrP, αSyn, and SOD1, can adopt an aberrant conformation that can template other pathological forms from the normal wildtype protein for seeding and trans-synaptic propagation throughout the brain [134,135]. αSyn is pathological in PD and MSA when it becomes nitrated, phosphorylated, and aggregated [136,137]; moreover, some post-translationally modified forms of αSyn cause prion-like spreading and disease in oligodendrocytes [135]. We found that human oligodendrocyte progenitors rapidly form prion-like forms of αSyn in response to NMDA receptor excitotoxicity. Clinical and animal studies of neonatal HI show that there is preferentially protracted brain network-wide degeneration [3,7,138,139]. Synapses and oligodendrocytes accumulating these aberrant proteins, as we have shown here, could be responsible for the connectome spreading and persistent damage in the neonatal brain with lifelong consequences, even after HT treatment.

## 5. Conclusions

This study shows that therapeutic HT shifts the neurodegeneration phenotype in human neonatal HIE but not in neonatal piglet models of HI where this therapy is strongly neuroprotective. The cooling and warming-induced phenotype shift in human neurons takes the form of ischemic necrosis being pushed to syncretic forms of necrosis and apoptosis. Such a change would be predicted by the cell death continuum concept [27]. We also introduce the possibility that prion-like proteinopathy could be a hub in the evolution of neonatal brain damage and could be a therapeutic target [140] for mitigating brain injury and chronic disability in infants and children, as it is for neurodegenerative diseases of aging. Oligomers and putative TCPs of αSyn, e.g., nitrated-Syn and aggregated αSyn, misfolded/oxidized SOD1, and PrP were detected with highly specific antibodies by immunohistochemistry, immunofluorescence, and/or immunoblotting in (1) human oligodendrocytes and neurons in cell cultures exposed to excitotoxicity; and (2) in human and piglet HIE brains. αSyn and SOD1 TCPs were seen in human HIE brains regardless of HT treatment, while in piglets HT mitigated their formation, suggesting that brain proteinopathy might titrate with injury severity.

Our study has limitations. One limitation is the inherent variability of human postmortem samples used experimentally. The variability can come from differences in clinical history (Table 1), postmortem interval, duration of brain immersion fixation, and brain cutting tissue sampling design (Appendix A). Each autopsy case represents a single time point that likely has unique features, limiting assessment of injury progression over time. Many cases need to be studied in attempts to comprehend the complexities of the cellular and molecular pathology of human HIE. Yet, the postmortem human HIE analysis is the gold standard that establishes what is needed in a preclinical experimental model to identify relevant cellular and molecular patho-mechanisms and therapeutic targets. Though deeply tragic, more autopsies should be requested by clinical neonatologists. Our neonatal human brain postmortem work is buttressed by our human oligodendrocyte and neuron cell models. Human iPS cell work has the potential to transform neonatal HIE research. The impact of our work in the piglet models could be hampered by species-specific activation or deactivation of molecular pathways [19] that may influence the translational applicability of the piglet model. However, this work shows for the first time that swine can form human-like TCPs in the brain without transgenic manipulations. We are intensely working on possible piglet–human differences now, as highlighted by the attenuation of TCP formation in piglets with HT but not in human HIE. Lastly, our approach was morpho-pathologically guided with protein-level validation by antibodies that detect abnormal pathological proteins that are used widely in adult neurodegenerative diseases. We did not address broad transcriptomic changes because detection of abnormal proteins provides a direct cytopathologic clue, and as shown here, sometimes with precise cellular domain resolution.

We do not yet have information on whether there is any brain clearance of these abnormal proteins with age, if the brain glymphatic system is compromised in experimental or clinical settings of neonatal HIE, and whether there are any relationships between the burden of abnormal protein accumulation in the brain and metrics on learning and memory. Our experimental piglet model can be used to identify relationships between neonatal insult severity and proteinopathy. These will be examined in the future using a 1-month survival cohort of HI piglets [141]. This work bridges neonatal brain injury and adult neurodegenerative disease and identifies IDPs and TCPs as new potential therapeutic targets for neonatal HIE that could be engaged by proteasome activators [67,142] and molecular tweezers [143,144].

## Figures and Tables

**Figure 1 cells-14-00586-f001:**
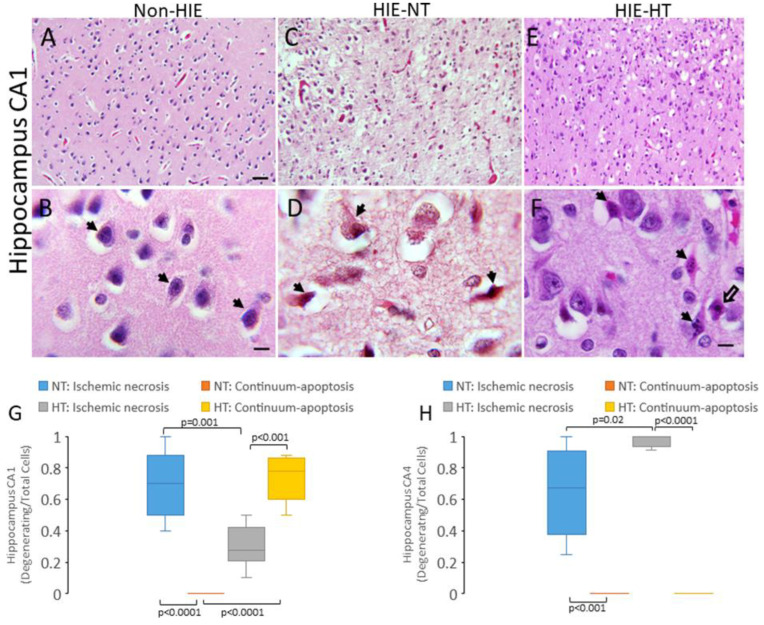
Hippocampal neuropathology in human infant HIE. Representative H&E staining of paraffin sections in cases of non-HIE (**A**,**B**), HIE without cooling (normothermia, NT, **C**,**D**), and HIE treated with hypothermia (HT, **E**,**F**) showing the CA1 region at low magnification (**A**,**C**,**E**) and at higher magnification for cellular detail (**B**,**D**,**F**). (**A**,**B**) In non-HIE cases, the neuropil is homogenously pink with many interspersed intact neuronal cell bodies (**B**, arrowheads). Scale bars: in (**A**) = 56 µm (same for **C**,**E**); in (**B**) = 18 µm (same for **D**); in (**F**) = 16 µm. (**C**,**D**) In HIE cases not treated with HT, the neuropil is pale, vacuolated and obliterated and the neuronal cell bodies show classic ischemic cell necrosis (**D**, arrowheads). (**E**,**F**) In HIE cases treated with HT, the neuropil is rescued, and the neuronal cell bodies are either intact or degenerating with non-necrotic morphologies consistent with either apoptosis-necrosis continuum cell death (arrowheads) or more apoptotic like (open arrow). (**G**) Quantitation of neurodegeneration in CA1 (HIE-NT *n* = 7; HIE-HT *n* = 10). Box plot showing mean values (with IQR and 5–95 percentile whiskers) of different structural forms of degenerating neurons (ischemic-necrotic or continuum-apoptosis). Cases of HIE-NT have predominantly necrotic neurodegeneration, while the neurodegeneration is shifted to continuum-apoptosis in HIE-cooled infants. (**H**) Quantitation of neurodegeneration in CA4 (intradentate). Box plot showing mean values (with IQR and 5–95 percentile whiskers) of different structural forms of degenerating neurons (ischemic-necrotic or continuum-apoptosis). Cases of HIE-NT and HIE-HT both have predominantly necrotic neurodegeneration in CA4 (HIE-NT *n* = 7; HIE-HT *n* = 10). Statistically significant *p* values are indicated.

**Figure 2 cells-14-00586-f002:**
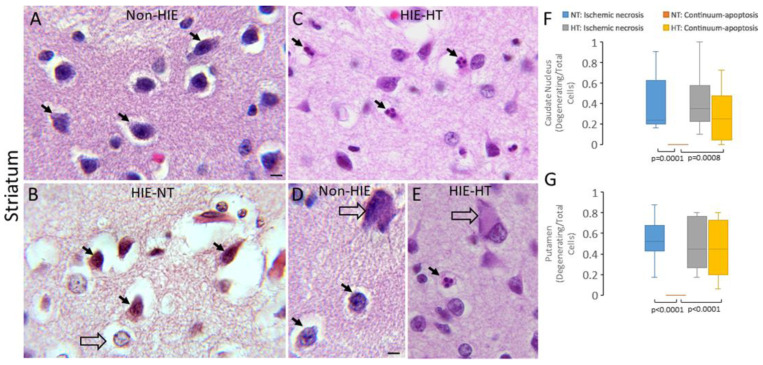
Striatal neuropathology in human infant HIE. Representative images are shown from H&E-stained paraffin sections of cases of non-HIE (**A**,**D**), HIE without cooling (normothermia, NT, **B**), and HIE treated with hypothermia (HT, **C**,**E**). (**A**) In non-HIE cases, the neuropil is homogenously pink with many interspersed intact round and ellipsoid neuronal cell bodies (arrows) and clear distinction between gray matter and penetrating white matter bundles. Scale bars: in (**A**) = 12 µm (same for **B**,**C**). (**B**) In HIE cases not treated with HT, the neuropil is pale and rarefied. The glial cells are swollen (open arrow). The neuronal cell bodies show classic ischemic cell necrosis (arrows) with exaggerated perineuronal space swelling. (**C**) In HIE cases treated with HT, the neuropil is rescued partially, and striatal neuronal cell bodies are either intact or degenerating with non-necrotic morphologies consistent with either apoptosis-necrosis continuum cell death or apoptosis variants (arrows). (**D**) In the non-HIE human infant striatum, the large striatal putative cholinergic interneurons (open arrow) are distinct from the principal medium-sized projection neurons (black arrows). Scale bar = 7 µm (same for **E**). (**E**) In HIE cases treated with HT the large striatal neurons are preserved but generally appear chromatolytic (open arrow) while their targets, the principal neurons, degenerate by either apoptosis-necrosis continuum cell death or apoptosis variants (arrow). (**F**,**G**) Quantitation of neurodegeneration in the human infant caudate nucleus (**F**) and putamen (**G**). Box plot showing mean values (with IQR and 5–95 percentile whiskers) of different structural forms of degenerating neurons (ischemic-necrotic or continuum-apoptosis). Cases of HIE-NT have predominantly necrotic neurodegeneration, while HIE-cooled infants have similar amounts ischemic-necrosis and continuum-apoptosis (HIE-NT *n* = 7; HIE-HT *n* = 10). Statistically significant *p* values are indicated.

**Figure 3 cells-14-00586-f003:**
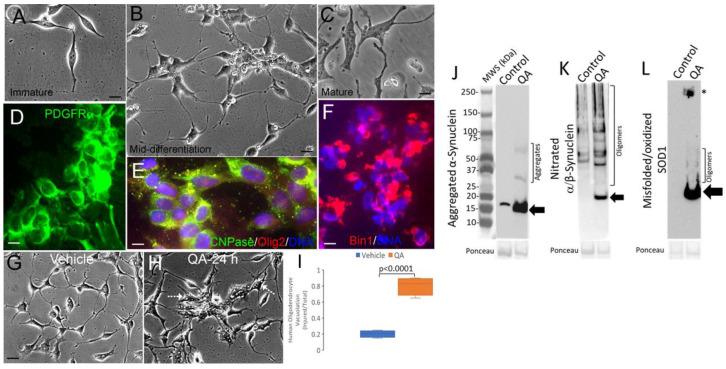
Directed differentiation of human induced pluripotent stem (iPS) cells into oligodendrocytes; their excitotoxicty is associated with formation of toxic conformer proteins (TCPs). (**A**–**C**) Live cell images showing the differentiation of human cells into healthy oligodendrocytes. Scale bars = 10 µm (**A**,**B**), 12 µm (**C**). (**D**–**F**) Immunophenotyping of human oligoprogenitors and oligodendrocytes with specific markers (PDGFR, CNPase, Olig2, Bin1). Scale bars = 10 µm (**D**,**F**), 15 µm (**E**). (**G**,**H**) Vehicle and QA (20 µM) exposures to mid-differentiated human oligodendrocytes. QA caused excitotoxicity evidenced by severe vacuolation (**H**, arrow). Scale bar (**G**) (same for **H**) = 20 µm. (**I**) Quantification of excitotoxic damage (cells with vacuolar pathology) to human oligoprogenitors. Box plot shows mean values (with IQR and 5–95 percentile whiskers. Three independent experiments were performed. (**J**–**L**) Western blots showing accumulation aggregated α-Syn (**J**), nitrated-Syn (**K**), and misfolded/oxidized SOD1 (**L**) in QA treated human oligoprogenitors. Black arrows identify monomers. Higher molecular weight aggregates or oligomers are indicated by bracket (**L**, asterisk identifies supramolecular weight aggregate of misfolded/oxidized SOD1). Ponceau stained membranes (bottom) show protein loading. Statistically significant *p* value is indicated.

**Figure 4 cells-14-00586-f004:**
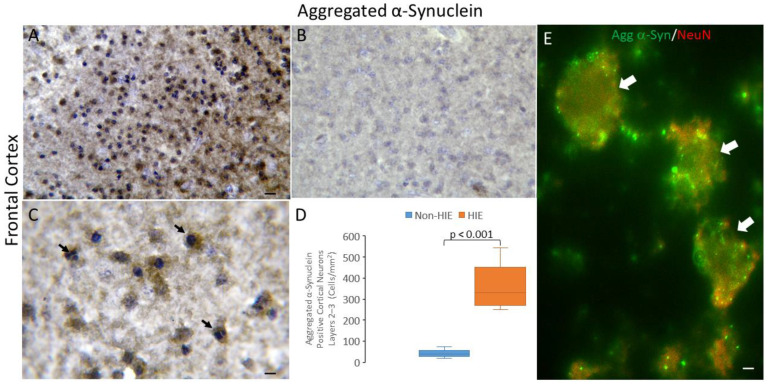
Synucleinopathy occurs in human neonatal HIE neocortex as demonstrated by the accumulation of aggregated α-Syn. (**A**) HIE case showing widespread accumulation of aggregated α-Syn immunoreactivity layers 2 and 3 of neocortex. Scale bar = 22 µm (same for **B**). (**B**) Non-HIE infants have scant aggregated α-Syn immunoreactivity in neocortex. (**C**) The aggregated α-Syn immunoreactivity in neonatal HIE neocortex accumulates in degenerating neurons with continuum cell death morphologies (black arrows). Scale bar = 10 µm. (**D**) Quantitation of the neuronal accumulation of aggregated α-Syn immunoreactivity in frontal cortical layers 2 and 3. Box plot showing mean values (with IQR and 5–95 percentile whiskers; non-HIE *n* = 8; HIE *n* = 14; 3 sections/case). Statistically significant *p* value is indicated. (**E**) Aggregated α-Syn colocalizes with subsets of cells positive for NeuN (white arrows). Some of the double-labeled neuron nuclei appear shrunken and attritional consistent with degenerating neurons. Scale bar = 2 µm.

**Figure 5 cells-14-00586-f005:**
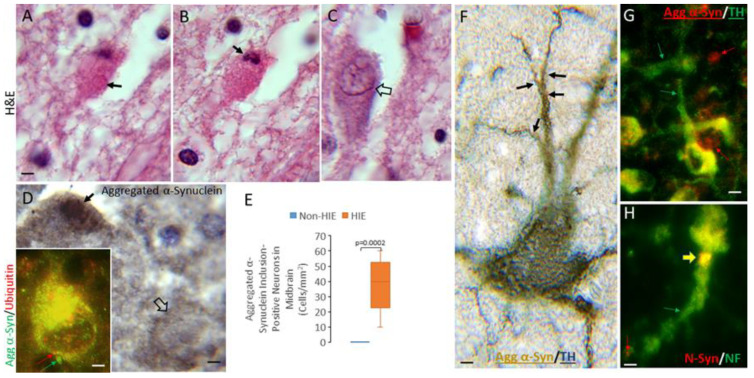
Synucleinopathy occurs in human neonatal HIE midbrain: nigral neurons accumulate aggregated α-Syn and form inclusions. (**A**,**B**) Substantia nigra neuron (at different focal planes in z-axis) showing continuum degeneration (appreciated by the chromatin aggregation pattern and intact cytoplasmic contour) and a cytoplasmic Lewy body inclusion. Scale bar in A = 8 µm (same for **B**,**C**). (**C**) Typical normal appearing substantia nigra neuron in a non-HIE case. (**D**) Aggregated α-Syn accumulates (immunoreactivity is brown from DAB) in the nucleus and cytoplasmic granules in some nigral neurons (black arrow), but in other nearby neurons, there is mainly faint cytoplasmic immunoreactivity (open arrow). Scale bar = 5 µm. Immunofluorescence (inset **D**) shows aggregated α-Syn (green) partly colocalizing (seen as yellow) with ubiquitin (red) in a large perinulcear apparent cytoplasmic accumulation distinct from the nucleus (pale region with non-overlapping red and green speckles). Red and green arrows identify single labeling for ubiquitin and aggregated α-Syn, respectively. Scale bar = 3 µm. (**E**) Quantitation of the neuronal accumulation of aggregated α-Syn immunoreactivity in substantia nigra neurons. Box plot showing mean values (with IQR and 5–95 percentile whiskers; non-HIE *n* = 8; HIE *n* = 12; 3 sections per case). Statistically significant *p* value is indicated. (**F**) Aggregated α-Syn (brown, black arrows) localizes to dendritic foci in morphologically normal appearing nigral dopaminergic neurons identified by tyrosine hydroxylase (TH, green-black, BDHC). Scale bar = 20 µm. (**G**) Immunofluorescent confirmation of the robust colocalization (seen as yellow) of TH-positive nigral neurons (green) with aggregated α-Syn (red) in an HIE case. Red and green arrows identify single labeling for aggregated α-Syn and TH, respectively. Scale bar = 10 µm. (**H**) Immunofluorescence for nitrated-synuclein (N-Syn) and neurofilament (NF) identifies focal accumulation of N-Syn within discrete domains of possibly swollen dendrites or axons (yellow arrow) and in the cell body. Red and green arrows identify single labeling for aggregated N-Syn and NF, respectively. Scale bar = 10 µm.

**Figure 6 cells-14-00586-f006:**
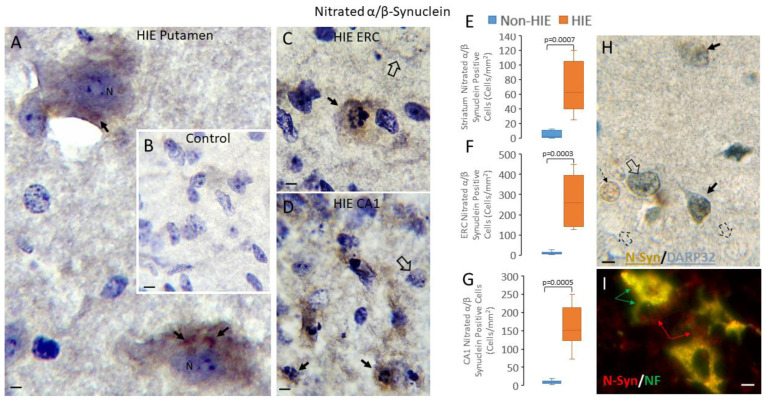
Synucleinopathy occurs in the human neonatal HIE brain as demonstrated by the accumulation of nitrated-Syn. (**A**) Putamen neurons in human HIE showing nitrated-Syn immunoreactivity (brown) in the cytoplasm and nucleus (N). Some of the nitrated-Syn forms cytoplasmic inclusions (arrows). CV counterstained. Scale bar = 4 µm. (**B**) Negative control section of HIE basal ganglia that was incubated with mouse IgG instead of mouse monoclonal nitrated-Syn antibody is blank. CV counterstained. Scale bar = 10 µm. (**C**) Neurons in entorhinal cortex (ERC) that are degenerating with a continuum cell death morphology (black arrow, shown by the chromatin clumping pattern in the nucleus) are positive for nitrated-Syn in cases of HIE. A nearby neuron (open arrow) does not have a degenerative phenotype and is negative for nitrated-Syn. Scale bar = 4 µm. (**D**) CA1 neurons degenerating with a continuum cell death signature are positive (black arrows) for nitrated-Syn in cases of HIE. A nearby neuron (open arrow) does not have a degenerative phenotype and is negative for nitrated-Syn. Scale bar = 7 µm. (**E**–**G**). Counts of positive nitrated-Syn neurons in striatum (**E**), ERC (**F**) and hippocampus CA1 (**G**) in HIE and non-HIE cases. Box plots show mean values (with IQR and 5–95 percentile whiskers; non-HIE *n* = 8; HIE *n* = 14; 3–4 sections/case). Statistically significant *p* values are indicated. (**H**) Colocalization of nitrated-Syn (brown) in DARP32-positive (blue-green) striatal neurons. Completely negative cells (hatched broad arrows). Nitrated-Syn single labeling (thin hatched arrow). DARP32 single labeling (open broad arrow). Double-labeled cells (solid black arrows). Scale bar = 10 µm. (**I**) Immunofluorescence confirmed the colocalization of nitrated-Syn (red) in NF68-positive (green) cortical neurons (seen as yellow) of entorhinal cortex. Nitrated-Syn single labeling (thin red arrows). NF68 single labeling (thin green arrows). Scale bar = 10 µm.

**Figure 7 cells-14-00586-f007:**
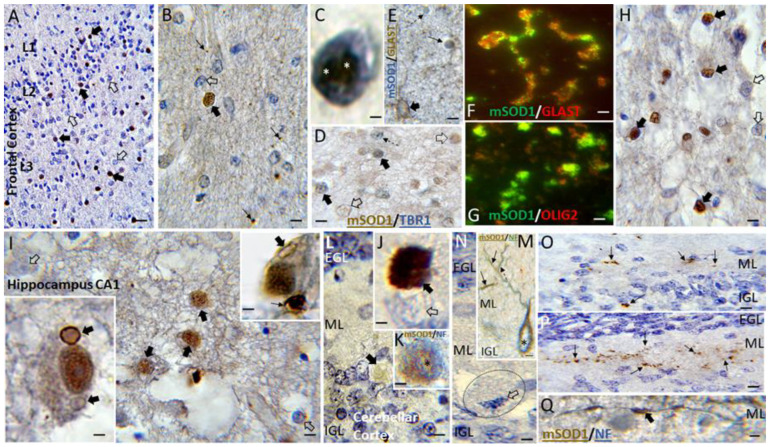
Misfolded/oxidized SOD1 accumulates in human neonatal HIE brain revealing nuclear, cytoplasmic, and possible connectome pathology. (**A**) Many cells are positive for misfolded/oxidized SOD1 (black arrows), but other cells are not positive (open arrows) in the HIE frontal cortex. Blue is CV counterstain. Shown is representative of *n* = 12 HIE cases and 3–4 sections from each brain region from each case. Scale bar = 56 µm. (**B**) Higher magnification shows the nuclear (black broad arrow) and neuropil (black thin arrows) localization of misfolded/oxidized SOD1. Some cells are negative (open arrow). The misfolded/oxidized SOD1 immunoreactivity in the neuropil is seen as granules and round structures (black thin arrows), possibly end stage apoptotic cells or degenerating terminals. Scale bar = 8 µm. (**C**) Misfolded/oxidized SOD1 immunoreactivity forms large nuclear inclusions in putative apoptotic cells in the cerebral cortex of neonatal HIE cases. Scale bar = 3 µm. (**D**) Colocalization of aberrant SOD1 (brown) in TBR1-positive (blue-green) cortical neurons. SOD1 single labeling (open broad arrow). TBR1 single labeling (hatched thin arrows). Double-labeled cells (solid black arrows). Scale bar = 10 µm. (**E**) Colocalization of aberrant SOD1 (blue-gray) in GLAST-positive (brown) cortical astrocytes. SOD1 single labeling (thin hatched arrow). Double-labeled normal appearing astrocyte (solid black broad arrow). Double-labeled astrocyte with condensed nucleus (solid black thin arrow). Scale bar = 10 µm. (**F**) Immunofluorescence confirms the colocalization of misfolded/oxidized SOD1 in astrocytes identified by GLAST (seen as yellow). Misfolded/oxidized SOD1 appeared compartmentalized within the cells. Scale bar = 5 µm. (**G**) Immunofluorescent colocalization of aberrant SOD1 (green) in Olig2-positive (red) cortical oligodendrocytes. Scale bar = 10 µm. (**H**) Misfolded/oxidized SOD1 is nuclear (black arrows) in the caudate nucleus but some cells are negative (open arrows). Scale bar = 8 µm. (**I**) Hippocampal CA1 neurons in HIE cases are positive for misfolded/oxidized SOD1 (black arrow), while some cells are completely negative (open arrow). CA1 neurons can form misfolded/oxidized SOD1 inclusions that are round and perinuclear (I, lower left inset, black arrows) or rectangular crystals in the cytoplasm (I, upper right inset, black arrow). Blue is CV counterstain. Scale bar = 6 µm (insets 10 µm). (**J**) Subsets of large cerebellar cells, possibly Purkinje cells, are strongly positive for misfolded/oxidized SOD1 (black arrow), while adjacent Purkinje cells were not positive (open arrow). Scale bar = 7 µm. (**K**) Colocalization of aberrant SOD1 (brown) in NF68-positive (blue-green) large Purkinje cells. Scale bar = 5 µm. (**L**) Degenerating end stage Purkinje cells (arrow) are faintly positive for misfolded/oxidized SOD1 while other cells in the external granule cell layer (EGL) molecular layer (ML), and internal granule cell layer are negative. Scale bar = 10 µm. (**M**) Aberrant SOD1 (brown, thin solid arrows) localizes to the dendritic foci of Purkinje cells (asterisk) identified by morphology and NF68 (green-black, BDHC). Other dendrite segments are negative for aberrant SOD1 (hatched thin arrows). Scale bar = 20 µm. (**N**) Purkinje cell that are undergoing classic ischemic necrotic cell death (open arrow, Purkinje cell is delineated by the elliptical hatched line). Scale bar = 8 µm. (**O**) Misfolded/oxidized SOD1-positive neuritic structures with a quasi-vertical arrangement reflective of passing climbing fibers in the granule cell layer and molecular layer can be seen in the HIE cerebellum. Scale bar = 10 µm. (**P**). Misfolded/oxidized SOD1-positive neurites can also have a horizontal organization passing through the molecular layer of cerebellum reminiscent of parallel fibers. Scale bar = 10 µm. (**Q**) Aberrant SOD1 (brown, black arrow) localizes to NF68-positive axons (blue-gray) in the cerebellar molecular layer (ML). Scale bar = 8 µm.

**Figure 8 cells-14-00586-f008:**
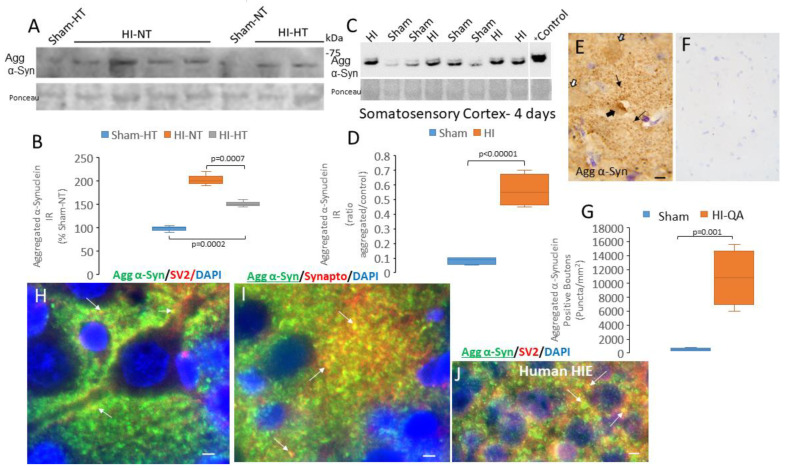
Synucleinopathy occurs in three different piglet models of HI brain damage as demonstrated by the accumulation of aggregated α-Syn. (**A**) Piglets were exposed to hypoxia–ischemia (HI) and treatment with hypothermia (HT) for 20 h followed by rewarming at 0.5 °C/hours or left at normothermia (NT) or treated with the sham procedure (with and without HT) and survived for 29 h. Western blotting of forebrain extracts show that aggregated α-Syn (oligomer-specific antibody Syn33) accumulated subacutely after HI. Ponceau S staining shows protein loading. (**B**) Graph of Western blot densitometry quantification of aggregated α-Syn in forebrain of sham-NT (*n* = 4), sham-HT (*n* = 4), HI-NT (*n* = 4), and HI-HT (*n* = 4) piglets. Box plot showing mean values (with IQR and 5–95 percentile whiskers). (**C**) Piglets were exposed to HI, but no hypothermia treatment, or were exposed to sham procedure and survived for 4 days. Western blotting of somatosensory cortex extracts shows that aggregated α-Syn (oligomer-specific antibody Syn33) accumulated days after HI. Brain sample from human mutant α-Syn-A53T transgenic mouse [85] was the ^+^control. Ponceau staining shows protein loading. (**D**) Graph of Western blot densitometry quantification for aggregated α-Syn at 4 days after HI (sham *n* = 4, HI *n* = 4). Box plot showing mean values (with IQR and 5–95 percentile whiskers). (**E**). Immunohistochemical localization of aggregated α-Syn in HI-QA piglets and 14 days after injury. Aggregated α-Syn was found enriched in attritional neurons in striatum (black arrow) while other neurons were lesser-enriched (gray arrow), and other neurons were negative (white arrow). Presynaptic bouton-like structures in the neuropil (thin arrows) were also positive. Scale bar = 10 µm (same for **F**). (**F**) Sham piglet neurons were largely negative for aggregated α-Syn suggesting that the antibody is detecting a pathological form of α-Syn. (**G**) Counts of positive aggregated α-Syn boutons in striatum of HI-QA and sham piglets at 15 days after injury (sham piglets *n* = 5, HI-QA piglets *n* = 6; 3 sections/piglet). Box plots show mean values (with IQR and 5–95 percentile whiskers). Statistically significant *p* values are indicated. (**H**) Immunofluorescence demonstrated that some aggregated α-Syn immunoreactivity (green) localizes to presynaptic terminals (white arrows, yellow) identified by SV2 (red) in the neuropil and in perineuronal/peridendritic regions of the HI piglet neocortex. Scale bar = 3 µm. (**I**) Immunofluorescence shows that aggregated α-Syn (green) colocalizes (white arrows, yellow) with synaptophysin (red) in the striatum of HI-QA piglets. Scale bar = 5 µm. (**J**) In a favorably preserved sample of the human HIE neocortex, immunofluorescence also showed that aggregated α-Syn (green) colocalized with SV2 (red) seen as yellow (white arrows). Some presynaptic terminals appeared swollen (left most white arrow). Scale bar = 6 µm.

**Figure 9 cells-14-00586-f009:**
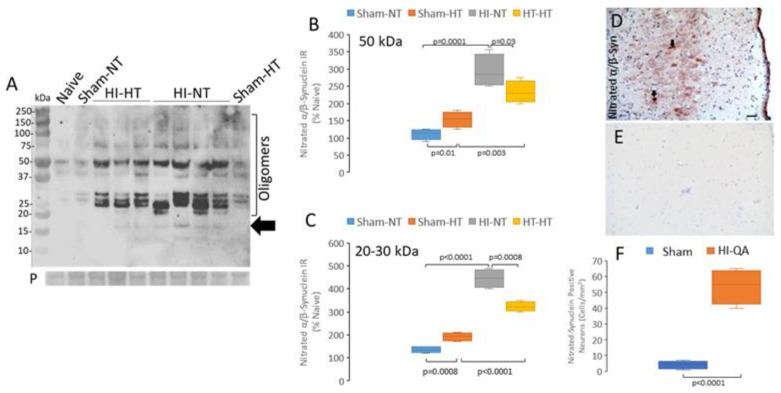
Synucleinopathy occurs in piglet HI brain as demonstrated by the accumulation of nitrated-Syn in two different models. (**A**) Piglets were exposed to hypoxia–ischemia (HI) and treatment with hypothermia (HT) for 20 h followed by rewarming at 0.5 °C/hours or left at normothermia (NT) or treated with sham procedure (with or without HT) and survived for 29 h. Western blotting of forebrain extracts show that nitrated-Syn (monoclonal antibody Syn12) accumulated subacutely after HI. Ponceau S staining shows protein loading. Arrow identifies the monomer form of Syn. (**B**) Graph of Western blot densitometry quantification of nitrated-Syn proteins at ~50 kDa in the forebrain of naïve (*n* = 4), sham-HT (*n* = 4), HI-NT (*n* = 4), and HI HT (*n* = 4) piglets. Box plot shows mean values (with IQR and 5–95 percentile whiskers). (**C**) Graph of Western blot densitometry quantification of nitrated-Syn proteins at ~20–30 kDa in forebrain of naïve (*n* = 4), sham-HT (*n* = 4), HI-NT (*n* = 4), and HI HT (*n* = 4) piglets. Box plot shows mean values (with IQR and 5–95 percentile whiskers). (**D**) Immunohistochemical localization of nitrated-Syn in HI-QA piglets and 14 days after injury. Nitrated-Syn was found enriched in layer 2 and 3 neurons in the entorhinal cortex (black arrows) while other neurons were lesser enriched. The neuropil was also positive. Scale bar = 24 µm (same for **E**). (**E**) Sham piglets were largely negative for nitrated-Syn confirming antibody specificity in detecting pathological forms of nitrated-Syn. (**F**) Counts of positive nitrated-Syn neurons in layer 2–3 of entorhinal cortex of HI-QA (*n* = 6) and sham (*n* = 5) piglets at 15 days after injury (3 sections/piglet brain). Box plots show mean values (with IQR and 5–95 percentile whiskers). Statistically significant *p* values are indicated.

**Figure 10 cells-14-00586-f010:**
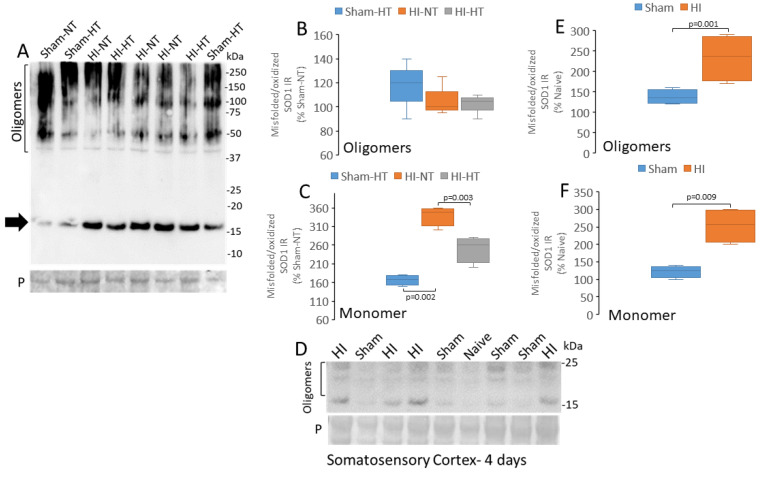
Misfolded/oxidized SOD1 accumulates in two different piglet models of HI. (**A**) Piglets were exposed to hypoxia–ischemia (HI) and treatment with hypothermia (HT) for 20 h followed by rewarming at 0.5 °C/hours or left at normothermia (NT) or treated with sham procedure (with or without HT) and survived for 29 h. Western blotting of forebrain extracts show that misfolded/oxidized SOD1 (monoclonal antibody C4F6) accumulated subacutely after HI. Ponceau S staining shows protein loading. Arrow identifies the modified monomer form of SOD1. (**B**) Graph of Western blot densitometry quantification of stable oligomers of misfolded/oxidized proteins at 37–250 kDa in the forebrain of sham-HT (*n* = 4), HI-NT (*n* = 4), and HI-HT (*n* = 4) piglets. Box plot shows mean values (with IQR and 5–95 percentile whiskers). (**C**) Graph of Western blot densitometry quantification of monomer of SOD1 at ~16 kDa in forebrain sham-HT, HI-NT, and HI HT piglets (4 piglets in each group). Box plot shows mean values (with IQR and 5–95 percentile whiskers). (**D**) Piglets were exposed to HI, but no hypothermia treatment, or were exposed to sham procedure and survived for 4 days. Western blotting of somatosensory cortex extracts shows that misfolded/oxidized SOD1 accumulated days after HI. Ponceau S staining shows protein loading. (**E**) Graph of Western blot densitometry quantification for misfolded/oxidized SOD1 oligomers at 4 days after HI (*n* = 4) or sham (*n* = 4) treatment. Box plot shows mean values (with IQR and 5–95 percentile whiskers). (**F**) Graph of Western blot densitometry quantification for misfolded/oxidized SOD1 monomer at 4 days after sham (*n* = 4) or HI (*n* = 4) treatments. Box plot shows mean values (with IQR and 5–95 percentile whiskers). Statistically significant *p* values are indicated. Data are based on three different Western blot experiments.

**Figure 11 cells-14-00586-f011:**
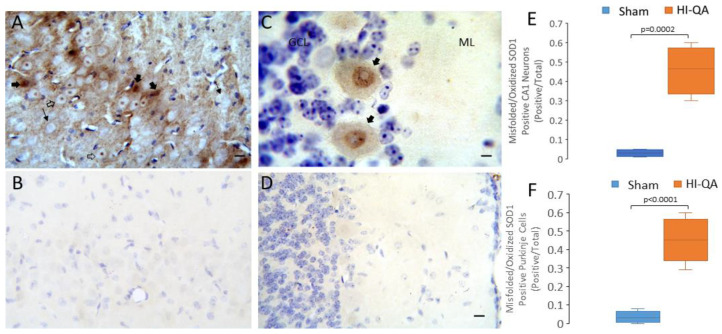
Aberrant misfolded/oxidized SOD1 accumulates selectively in subsets of neurons in HI piglet brain. (**A**,**B**) Hippocampus of HI-QA pig at 14 days after injury showing positivity for misfolded/oxidized SOD1 (C4F6 monoclonal antibody) in CA1 pyramidal neuron cell bodies (**A**, black arrows) and surrounding neuropil. Other neurons have only nucleolar SOD1 (**A**, open arrow) and some neurons have none or only cytoplasmic SOD1 labeling (**A**, thin arrows), while sham piglet CA1 shows essentially no C4F6 positivity (**B**). Scale bar in (**A**) = 17.5 µm (same for **B**). (**C**,**D**) Cerebellar cortex of HI-QA pig at 14 days after injury showing positivity for misfolded/oxidized SOD1 in Purkinje cell bodies (**C**, black arrows) with nuclei enriched in aberrant SOD1; some neuropil immunoreactivity is in the molecular layer (ML) and no labeling in the granule cell layer (GCL). Sham piglet cerebellar cortex shows only faint C4F6 positivity (**D**). Scale bars = 7.5 µm (**C**); 17.5 µm (**D**). (**E**) Box plot showing mean values (with IQR and 5–95 percentile whiskers) of misfolded/oxidized SOD1-positive neuron cell bodies in CA1 in HI-QA piglets (*n* = 6) at 15 days after injury or piglets with sham treatment (*n* = 5), and 3 brain sections/piglet. (**F**) Box plot showing mean values (with IQR and 5–95 percentile whiskers) of misfolded/oxidized SOD1-positive Purkinje cells in HI-QA piglets (*n* = 6) at 15 days after injury or sham treatment (*n* = 5). Statistically significant *p* values are indicated.

**Figure 12 cells-14-00586-f012:**
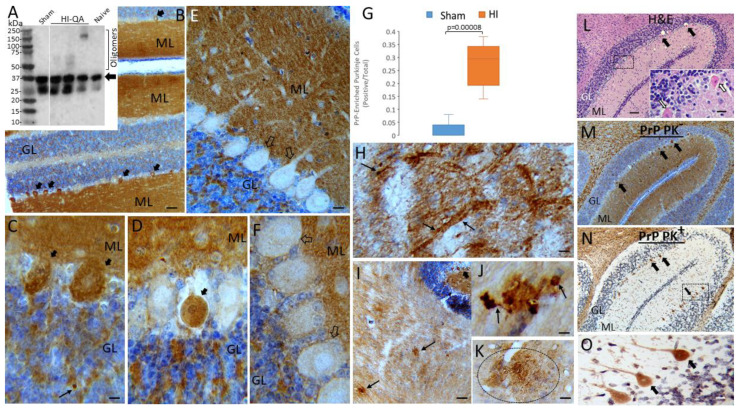
Prion protein (PrP) abnormalities emerge in two different piglet models of HI. (**A**) Western blot demonstrating PrP antibody specificity, with monomer (arrow) and immature forms below, in naïve and sham piglet brain and showing accumulation of PrP oligomers in the somatosensory cortex at 48 h after HI-QA injury. Representative of three different experiments. (**B**–**D**). Localization of PrP in HI piglet cerebellar cortex at 4 days after asphyxic cardiac arrest. Panoramic low magnification view (**B**) showing enrichment of PrP immunoreactivity in the molecular layer (ML) and in some Purkinje cells (**B**, black arrows), while other Purkinje cell are not positive. Higher magnification images of HI piglet cerebellum (**C**,**D**) showing PrP accumulation in Purkinje cells that are degenerating with cytoplasmic vacuolation (**C**, black arrows) or attrition consistent with the cell death continuum (**D**, black arrow). Putative granule cells (C, black arrow) die apoptotically. Scale bars = 70 µm (**B**), 12.5 µm (**C**, same for **D**,**F**). (**E**,**F**) In sham control piglet cerebellum, Purkinje cell bodies are generally PrP-negative (**C**, open arrows) or have only faint cytoplasmic PrP immunoreactivity. Scale bars = 40 µm (**E**). (**G**) Box plot showing mean values (with IQR and 5–95 percentile whiskers) of PrP-positive Purkinje cell bodies in cerebellar cortex in HI piglets (*n* = 6) at 4 days after injury or piglets with sham treatment (*n* = 6). Data are based on three brain sections/piglet. (**H**) In deep cerebellar nuclei of HI piglets, PrP-positive dystrophic axons and swollen peridendritic axon terminals are present (arrows). Scale bar = 28 µm. (**I**) PrP-positive plaque-like lesions are also present in the cerebellar deep nuclei and surrounding white matter (arrows). Abnormal PrP-laden Purkinje cell bodies are seen (black arrow, upper right). Scale bar = 56 µm. (**J**) Some PrP-positive plaques are neuritic (arrows). Scale bar = 10 µm. (**K**) Some PrP-positive plaques are diffuse (hatched ellipse). Scale bar = 6 µm. Statistically significant *p* value is indicated. (**L**) H&E staining of piglet cerebellar section at 4 days after HI. Eosinophilic ischemic necrotic Purkinje neurons are identified (black arrows) between the molecular layer (ML) and the granule cell layer (GL). Hatched box shown as inset with Purkinje neurons (black arrows) with eosinophilic cytoplasm and pyknotic nucleus. Scale bars = 40 µm (inset 15 µm). (**M**) Adjacent section not treated with proteinase K (PK) and stained for PrP showing enrichment of immunoreactivity in the molecular layer (ML), isolated Purkinje cell bodies (black arrows), granule cell layer (GL), and subcortical white matter. Scale bar = 40 µm (same for **N**). (**N**) Adjacent section treated with PK and stained for PrP showing near complete digestion of PrP in the ML, GL, and infra-GL white matter, but isolated Purkinje cell bodies (black arrows) remain strongly positive. (**O**) Hatched box in N showing PK-resistant Purkinje cell PrP immunoreactivity (black arrows). Scale bar = 10 µm.

**Table 1 cells-14-00586-t001:** Human Infant Autopsy Cases Used.

Case Identifier	Age at Birth (Weeks)	Putative Clinical Insult	Apgar (When at Birth)	Therapeutic Hypothermia	Last Recorded Blood pH	Age (at Death)	Postmortem Delay (Hours)
A15-7	36 (37)	Fetal deceleration and emergency C-section. Resuscitation involving chest compressions, gas of 6.9, and clinical seizures	1 (1 min)	Yes	7.283 arterial; 7.251 capillary	7 days	24
A16-13	41.6 (42.9)	Shoulder dystocia; prolonged ruptured membranes	1, 1, 5 (1, 5 and 10 min)	Yes	7.425 arterial; 7.259 capillary	9 days	12
A16-30	34 (34.3)	Car collision, placental abruption, emergency C-section, DIC in newborn, neonatal respiratory failure	1, 1 (1 and 5 min)	Yes	7.017 capillary; 6.971 arterial	2 days	144
A17-1	39.3 (39.9)	Rupture of membranes, C-section, required chest compressions, cord gas of 6.9	1, 3, 4 (1, 5, and 10 min)	Yes	7.221 arterial	4 days	12
A17-3	38 (38.4)	Uncontrolled insulin-dependent diabetes (mother), emergency C-section, perinatal asphyxia, chest compressions	0, 1, 3 (1, 5, and 10 min)	Yes	7.271 arterial; 7.279 capillary	3 days	24
A17-14	24 (48.6)	Chronic lung disease with secondary pulmonary hypertension, hypoxemic respiratory failure, worsening hypotension	8, 9 (1 and 5 min)	0 (missed therapeutic time window)	7.439 arterial; 7.317 capillary	172 days	25
A18-2	35.3 (35.9)	Diamniotic dichorionic twins, premature rupture of membranes, emergency C-section, Traumatic delivery (cephalohematoma, focal subdural hemorrhage, subarachnoid hemorrhages), chest compressions	1, 0, 0 (1, 5, and 10 min)	Yes	7.459 arterial; 7.283 capillary	4 days	16
A18-3	39.9 (40.4)	Secondary apnea, respiratory failure, multiorgan failure, possible septic shock	5, 2 (1, and 5 min)	Yes	7.312 arterial	4 days	24
A18-17	34.3 (36.4)	HIV+ mother via emergency C section for fetal deceleration for systole, chest compressions	0, 0, 1 (1, 5, and 10 min)	0 (missed therapeutic time window)	7.217 arterial; 7.391 capillary	15 days	48
A18-28	40.3 (40.4)	Non-reassuring fetal heart rate, acute phlebitis of umbilical cord, acute chorioamnionitis of membranes, chorionic plate had acute subchorionitis, tight nuchal chord, placental SGA	0, 0, 1 (1, 5, and 10 min)	Yes	6.6; BD > 20	1 day	408
4314	38	Respiratory insufficiency, anoxic encephalopathy	NA	No	NA	4 days	21
667	38	Acute cardiac arrhythmia-arrest (Non-HIE control)	No	No	NA	353 days	13
828	36	Meconium aspiration, seizure disorder	8, 9	No	NA	90 days	10
731	36	Severe birth anoxia, anoxic encephalopathy, seizure disorder	NA	No	NA	360 days	14
A54802	39	Rupture of membranes, cardiorespiratory arrest	1, 0, 0, 2, 3 (1, 5, 10, 15, 20 min)	Yes	NA	4 days	24
A56447	35	Non-reassuring fetal statue, C-section, immediate apnea, respiratory arrest. Anoxic encephalopathy	2, 3, 3 (1, 5, 10)	No (missed therapeutic time window)	NA	5 days	24
A54854	34.5	Maternal gestational diabetes, late decelerations during cerclage removal, emergency C-section	1, 1, 1, 1, 2 (1, 5, 10, 15, 20)	Yes	NA	1 day	22
A54550	36	SMA (Non-HIE control)	NA	NA	NA	14 days	18
4358	39	Non-HIE control, accidental death	8, 9	NA	NA	9 days	25
4360	40	Non-HIE control, accidental death	8, 9	NA	NA	202 days	33
4361	40	Non-HIE control, undetermined	8, 9	NA	NA	236 days	27
4388	40	Non-HIE control, accidental death	8, 9	NA	NA	147 days	53
4389	34	Non-HIE control, accidental death	8, 9	NA	NA	79 days	27
4415	40	Non-HIE control, undetermined	8, 9	NA	NA	146 days	46
4418	39	Non-HIE control, accidental death	8, 9	NA	NA	76 days	24
4421	39	Non-HIE control, accidental death	8, 9	NA	NA	84 days	40
4460	39	Non-HIE control, accidental death	8, 9	NA	NA	21 days	25
5947	40	Non-HIE control, accidental death	8, 9	NA	NA	179 days	11

**Table 2 cells-14-00586-t002:** Piglet Encephalopathy Models Studied.

	Injury/Insult	Survival and Group Sizes	Experimental Use	Justification
Piglet (2–3 days old male)	Global hypoxia-ischemia (HI) or sham with normothermic (NT) or hypothermic (HT) recovery	2–7 days. Sham-NT (*n* = 6), Sham-HT (*n* = 10), HI-NT (*n* = 8), HI-HT (*n* = 10)	Histology: H&E staining for neuronal counting and cell death morphology	Clinically relevant with therapeutic; Survival can be limited by seizures.
Piglet (2–4 days old male)	Global hypoxia-ischemia (HI) with NT or HT recovery	29 h. Sham-NT (*n* = 4), Sham-HT (*n* = 4), HI-NT (*n* = 4), HI-HT (*n* = 4), naïve (*n* = 4)	Western blotting for toxic conformer proteins	Clinically relevant with therapeutic; short survival to avoid seizures.
Piglet (2–3 days old male)	Global HI (no temperature management)	96 h (4 days). Sham (*n* = 6), HI (*n* = 6)	Histology: immunohistochemistry and immunofluorescence	Gold standard historical model. Much information on neuropathology without cooling and need to control for cooling effects on brain. Ideal balance of survival time and requirement for animal care.
Piglet (2–3 days old male)	Global HI (no temperature management)	96 h (4 days). Sham (*n* = 4), HI (*n* = 4)	Western blotting for toxic conformer proteins	As above
Piglet (2–3 days old male)	Global HI plus quinolinic acid (QA) excitotoxic lesion (2-hit protocol, no temperature management)	48 h. Sham (*n* = 4), HI-QA (*n* = 4), Vehicle (*n* = 4), naïve (*n* = 4)	Western blotting for toxic conformer proteins	Newest model that bridges in vivo and human cell culture QA experiments. Exquisite regionally specific white and gray matter molecular profiling attractive.
Piglet (2–3 days old male)	Global HI plus quinolinic acid (QA) excitotoxic lesion (2-hit protocol, no temperature management)	15 days. Sham (*n* = 5), HI-QA (*n* = 6)	Histology: immunohistochemistry and immunofluorescence	Newest model that capitalizes on lesser asphyxic heart damage for longer survival without need to control for cooling. Short and long-term survival attractive.

**Table 3 cells-14-00586-t003:** Antibodies to Toxic Conformer Proteins Used.

Antibody	Characterization	Target/Antigen	IgG Type	Source
α-Synuclein (Syn) Aggregate, clone MJFR-14-6-4-2	[69,70]	α-Syn conformation-specific/full-length α-Syn	Rabbit monoclonal	Abcam, Waltham, MA, USA
α-Syn Aggregate, clone 5G4	[71,72]	Aggregated α-Syn/KLH-conjugated peptide corresponding to human aggregated α-Syn	Mouse monoclonal	Millipore-Sigma, St. Louis, MO, USA
α-Syn Oligomer (Syn33)	[73]	Wildtype full-length α-Syn oligomers	Rabbit polyclonal	Millipore-Sigma
Nitrated Syn, clone Syn12	[57]	Nitrated Syn/α-Syn nitrated at Tyr125 and Tyr136 (β-Syn nitrated at Tyr130)	Mouse monoclonal	Santa Cruz Biotechnology, Dallas, TX, USA
Misfolded-Aggregated SOD1, clone C4F6	[74]	SOD1/full-length SOD1 apoenzyme	Mouse monoclonal	Medimabs, Montreal, Quebec, Canada
Misfolded-Aggregated SOD1, clone B8H10	[75]	SOD1/full-length SOD1 apoenzyme	Mouse monoclonal	Medimabs
PrP, clone F89/160.1.5	[76]	PrP n-IHFG-n	Mouse monoclonal	Invitrogen-ThermoFisher Scientific, Waltham, MA, USA

**Table 4 cells-14-00586-t004:** Antibodies Used for Cell Type and Synapse Identification in Combination with Antibodies to Toxic Conformer Proteins.

Cell Type Target	Identification Purpose	IgG Type	Source
Neurofilament 68 (NF68)	Neuron cell body and axon	Rabbit polyclonal	Abcam Ab9035
NeuN	Neuron cell body	Rabbit polyclonal	Millipore ABN78
Ubiquitin	Inclusions	Mouse monoclonal	Abcam Ab7254
CNPase	Oligodendrocytes	Mouse monoclonal, clone 11-5B	Millipore-Sigma
Olig2	Oligodendrocytes	Rabbit monoclonal, EPR2673	Abcam, Ab109186
GLAST	Astrocytes	Rabbit polyclonal	Proteintech, 20785-1-AP
Tyrosine hydroxylase	Midbrain dopaminergic neurons	Rabbit polyclonal	Novus Biologicals, NB300-109
DARP32	Striatal neurons	Rabbit monoclonal, EP720Y	Abcam Ab40801
TBR1	Cortical neurons	Rabbit polyclonal	Abcam Ab31940
Synaptophysin	Presynaptic terminals	Mouse monoclonal. 4E12C4	Proteintech, 67864-1-Ig
SV2a	Presynaptic terminals	Rabbit polyclonal	Synaptic Systems, Gottingen, Germany, 119003
Synapsin 1 and 2	Presynaptic terminals	Rabbit polyclonal	Synaptic Systems, 106002
Cysteine String Protein (CSP)	Presynaptic terminals	Rabbit polyclonal	Stressgen Bioreagents, Victoria, British Columbia, Canada, VAP-SV003
Munc18	Presynaptic terminals	Mouse monoclonal, 31/munc-18	BD Transduction Laboratories, San Diego, CA, USA, 610336
β-synuclein	Neurons and presynaptic terminals	Rabbit monoclonal, EP1537Y	Epitomics, Burlingame, CA, USA, 1977-1

## Data Availability

All data related to these experiments are available by contacting Lee J. Martin.

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
