# Peer review of "Hypothermia Shifts Neurodegeneration Phenotype in Neonatal Human Hypoxic–Ischemic Encephalopathy but Not in Related Piglet Models: Possible Relationship to Toxic Conformer and Intrinsically Disordered Prion-like Protein Accumulation"

_cells, 2025, doi:10.3390/cells14080586_

Round 1

Reviewer 1 Report

Comments and Suggestions for Authors

In the article "Hypothermia Shifts Neurodegeneration Phenotype in Neonatal Human Hypoxic-Ischemic Encephalopathy but not in Related Piglet Models: Possible Relationship to Toxic Conformer and Intrinsically Disordered Prion-Like Protein Accumulation", the authors tested the hypothesis that controlled hypothermia partially rescues cell death progression after hypoxic-ischemic damage in human neonates and in piglet models. In addition the authors tested the possibility that proteinopathies were present in cultured human oligodendrocytes and neurons subject to neurotoxic stress with quinolinic acid.

The introduction is written clearly, and provides a comprehensive review of published results that support the articulation of the main hypothesis tested in the study.

The materials and methods section explain thoroughly the approach and different assays used for the human, animal model, and cell culture components of the study.

The results section contains 12 figures and 2 tables. Figures show high quality exemplar images from human and animal model samples and quantification of observed features from the images. When relevant, figures include western blot pictures to illustrate biochemical changes associated with relevant proteinopathy such as synuclein, and SOD1 and PrP in cell cultures and tissue samples. This is a notable highlight of the study, and provides evidence that proteinopathies observed in the context of adult aging neurodegenerative pathology can be observed in the context of hypoxic-ischemic damage in neonates.

The discussion section contains a critical view of the experimental design and the implications of the observations in different contexts (human, animal model and cell culture). Overall the conclusions are supported by the results. The study should be of interest to a brioad audience, including researchers interested in mitochondrial stress in the context of hypoxic-ischemic pathology, who should reconsider the important effects of neurotoxic damage.

Author Response

Comment:
In the article "Hypothermia Shifts Neurodegeneration Phenotype in Neonatal Human Hypoxic-Ischemic Encephalopathy but not in Related Piglet Models: Possible Relationship to Toxic Conformer and Intrinsically Disordered Prion-Like Protein Accumulation", the authors tested the hypothesis that controlled hypothermia partially rescues cell death progression after hypoxic-ischemic damage in human neonates and in piglet models. In addition, the authors tested the possibility that proteinopathies were present in cultured human oligodendrocytes and neurons subject to neurotoxic stress with quinolinic acid.

The introduction is written clearly and provides a comprehensive review of published results that support the articulation of the main hypothesis tested in the study. The materials and methods section explains thoroughly the approach and different assays used for the human, animal model, and cell culture components of the study.

The results section contains 12 figures and 2 tables. Figures show high quality exemplar images from human and animal model samples and quantification of observed features from the images. When relevant, figures include western blot pictures to illustrate biochemical changes associated with relevant proteinopathy such as synuclein, and SOD1 and PrP in cell cultures and tissue samples. This is a notable highlight of the study and provides evidence that proteinopathies observed in the context of adult aging neurodegenerative pathology can be observed in the context of hypoxic-ischemic damage in neonates.

The discussion section contains a critical view of the experimental design and the implications of the observations in different contexts (human, animal model and cell culture). Overall, the conclusions are supported by the results. The study should be of interest to a broad audience, including researchers interested in mitochondrial stress in the context of hypoxic-ischemic pathology, who should reconsider the important effects of neurotoxic damage.

Response: We sincerely thank Reviewer 1 for the thorough and supportive review. We are especially grateful for the recognition of the manuscript’s clarity, methodological rigor, and translational relevance. We appreciate the thoughtful remarks on our integration of human and experimental models, the quality of our figures and biochemical validation, and the broader implications of our findings. We are encouraged that the conclusions were found to be well supported and that the study may serve as a reference for future work in the field. We have made minor edits for clarity and added a paragraph to the “Discussion” section addressing study limitations, as also requested by Reviewer 2.

Reviewer 2 Report

Comments and Suggestions for Authors

This manuscript presents a comprehensive and rigorous study exploring the similarities and differences in the therapeutic effects (or lack thereof) of therapeutic hypothermia in hypoxic-ischemic encephalopathy (HIE) in human subjects, widely used animal models and cell lines. The study delves into variations in neuronal death phenotypes and provides compelling evidence implicating proteinopathy in the observed neurodegenerative patterns.

Beyond the meticulous investigation conducted, two key strengths of this manuscript distinguish it as a valuable contribution to the field. First, the breadth of materials analyzed enhances the study's translational relevance. The authors include samples from human subjects, arguably the most clinically pertinent cohort, alongside multiple widely used piglet models and human cell lines that have not undergone oncogenic immortalization. The use of these primary cell lines ensures that the results are more physiologically relevant compared to studies relying on immortalized cell lines. This makes the study a potential reference point for future research examining the therapeutic efficacy of interventions targeting HIE in piglet models or in vitro systems.

Second, the extensive and transparent reporting on antibody use is commendable. Antibody specificity and reliability are critical for ensuring reproducibility in studies detecting proteinopathic changes. The detailed documentation provided in this manuscript offers a valuable resource for future research, helping to validate and optimize detection methods while minimizing potential artifacts arising from non-specific antibody interactions.

Minor Comments:

The manuscript describes that in cooled infants, the predominant neurodegenerative phenotype in the forebrain morphologically resembles a hybrid or continuum of necrotic and apoptotic characteristics rather than presenting as distinct, independent cell death pathways. Would the authors consider this phenomenon consistent with necroptosis? If so, the inclusion of necroptosis markers could further strengthen the argument for a continuum of cell death and may also provide potential biomarkers to assess treatment efficacy.

The detection of proteinopathic hallmarks is particularly intriguing. Given this, have the authors considered investigating markers of autophagic activity and/or chaperone protein expression? Protein aggregation and the accumulation of misfolded proteins often result from an overwhelmed or dysfunctional autophagic response. Exploring these pathways could provide mechanistic insights into whether autophagy impairment contributes to the observed proteinopathy and cellular stress.

A broad transcriptomic analysis of the samples could offer additional valuable insights. Such an approach might identify key differences and similarities in the molecular mechanisms underlying HIE pathogenesis and hypothermia-induced neuroprotection across the studied models.

Finally, what are the key limitations of this study? A discussion of potential confounding factors, methodological constraints, or avenues for future research would further enhance the manuscript’s impact.

Author Response

Comment:
This manuscript presents a comprehensive and rigorous study exploring the similarities and differences in the therapeutic effects (or lack thereof) of therapeutic hypothermia in hypoxic-ischemic encephalopathy (HIE) in human subjects, widely used animal models, and cell lines. The study delves into variations in neuronal death phenotypes and provides compelling evidence implicating proteinopathy in the observed neurodegenerative patterns.

Beyond the meticulous investigation conducted, two key strengths of this manuscript distinguish it as a valuable contribution to the field. First, the breadth of materials analyzed enhances the study's translational relevance. The authors include samples from human subjects, arguably the most clinically pertinent cohort, alongside multiple widely used piglet models and human cell lines that have not undergone oncogenic immortalization. The use of these primary cell lines ensures that the results are more physiologically relevant compared to studies relying on immortalized cell lines. This makes the study a potential reference point for future research examining the therapeutic efficacy of interventions targeting HIE in piglet models or in vitro systems.

Second, the extensive and transparent reporting on antibody use is commendable. Antibody specificity and reliability are critical for ensuring reproducibility in studies detecting proteinopathic changes. The detailed documentation provided in this manuscript offers a valuable resource for future research, helping to validate and optimize detection methods while minimizing potential artifacts arising from non-specific antibody interactions.

Reviewer 2 Minor Comments

Comment 1 (1st paragraph):
The manuscript describes that in cooled infants, the predominant neurodegenerative phenotype in the forebrain morphologically resembles a hybrid or continuum of necrotic and apoptotic characteristics rather than presenting as distinct, independent cell death pathways. Would the authors consider this phenomenon consistent with necroptosis? If so, the inclusion of necroptosis markers could further strengthen the argument for a continuum of cell death and may also provide potential biomarkers to assess treatment efficacy.

Response:
We agree that the hybrid morphology observed could involve regulated necrotic mechanisms such as necroptosis. We have published in this regard in rodent HI models (https://doi.org/10.1038/jcbfm.2010.72). Your idea is good. We are pursuing this ideal. A problem is that for several of the key players for necroptosis, including RIP1, RIP3, and cleaved caspase-8, we have not identified antibodies that we are comfortable using on human and pig samples. Our analysis of RIPs and caspase-8 need more time for antibody verification. In addition, we respectfully believe that the inclusion of necroptosis markers would not meaningfully advance the central narrative of this study, which centers on morphological divergence and proteinopathy across models. Given the irreplaceable nature of human postmortem tissue, we aim to preserve these samples for analyses most closely aligned with our defined hypotheses using well characterized antibody tools. We view necroptosis as a promising future direction and appreciate the reviewer highlighting its relevance.

Comment 2 (2nd paragraph):
The detection of proteinopathic hallmarks is particularly intriguing. Given this, have the authors considered investigating markers of autophagic activity and/or chaperone protein expression? Protein aggregation and the accumulation of misfolded proteins often result from an overwhelmed or dysfunctional autophagic response. Exploring these pathways could provide mechanistic insights into whether autophagy impairment contributes to the observed proteinopathy and cellular stress.

Response:
We appreciate the reviewer’s thoughtful  and logical suggestion. We agree that the balance of protein folding and clearance mechanisms, including autophagy, is central to proteostasis. Autophagy is very difficult to assess convincingly in static tissue sections, particularly without electron microscopy. Moreover, we do not have human autopsy tissue for western blotting. Our focus in the current study was to demonstrate the presence of toxic protein conformers and prion-like pathology in neonatal HIE using translational markers. Inclusion of autophagy and chaperone markers, while relevant, would necessitate additional use of precious tissue, alternative tissue protocols and tissue preparations, and would not substantively alter our mechanistic conclusions. We consider this an important avenue for future work designed specifically to investigate clearance pathways.

Comment 3 (3rd paragraph):
A broad transcriptomic analysis of the samples could offer additional valuable insights. Such an approach might identify key differences and similarities in the molecular mechanisms underlying HIE pathogenesis and hypothermia-induced neuroprotection across the studied models.

Response:
We agree that RNA sequencing can serve as a powerful hypothesis-generating tool. Indeed, we considered transcriptomic profiling in the early stages of the study. Transcriptomic analysis in our pig models is difficult because of the paucity of annotated databases. For human HIE, we first wanted to do a comprehensive neuropathological review of the human samples. To this end, we will adopt a targeted strategy informed by clear and reproducible morphological phenotypes. This allowed us to directly test our hypotheses and validate observations at the translational level while preserving tissue resources. We anticipate using RNA sequencing in future studies designed specifically for broader discovery-phase analysis.

Comment 4 (4th paragraph):
Finally, what are the key limitations of this study? A discussion of potential confounding factors, methodological constraints, or avenues for future research would further enhance the manuscript’s impact.

Response:
We thank the reviewer for this excellent suggestion. We have added content in the revised Conclusions that acknowledges several key limitations. These include the inherent variability of human postmortem samples, including differences in clinical history and postmortem interval; the use of single time points, which limits assessment of injury progression over time; and the species-specific responses that may influence the translational applicability of the piglet model. We also note that while our approach focused on morphologically guided, protein-level validation, it does not capture broader transcriptomic changes. These limitations and their implications for future research are now clearly stated in the manuscript.